

# CFTs with U(m) × U(n) global symmetry in 3D and the chiral phase transition of QCD

Stefanos R. Kousvos[1] and Andreas Stergiou[2]

**1** Department of Physics, University of Pisa and INFN, Largo Pontecorvo 3, I-56127 Pisa, Italy
**2** Department of Mathematics, King's College London, Strand, London WC2R 2LS, United Kingdom

## Abstract

Conformal field theories (CFTs) with $U(m) \times U(n)$ global symmetry in $d = 3$ dimensions have been studied for years due to their potential relevance to the chiral phase transition of quantum chromodynamics (QCD). In this work such CFTs are analyzed in $d = 4-\varepsilon$ and $d = 3$. This includes perturbative computations in the $\varepsilon$ and large-$n$ expansions as well as non-perturbative ones with the numerical conformal bootstrap. New perturbative results are presented and a variety of non-perturbative bootstrap bounds are obtained in $d = 3$. Various features of the bounds obtained for large values of $n$ disappear for low values of $n$ (keeping $m < n$ fixed), a phenomenon which is attributed to a transition of the corresponding fixed points to the non-unitary regime. Numerous bootstrap bounds are found that are saturated by large-$n$ results, even in the absence of any features in the bounds. A double scaling limit is also observed, for $m$ and $n$ large with $m/n$ fixed, both in perturbation theory as well as in the numerical bootstrap. For the case of two-flavor massless QCD existing bootstrap evidence is reproduced that the chiral phase transition may be second order, albeit associated to a universality class unrelated to the one usually discussed in the $\varepsilon$ expansion. Similar evidence is found for the case of three-flavor massless QCD, where we observe a pronounced kink.



# 1   Introduction

The Lagrangian of quantum chromodynamics (QCD) with $N_f$ massless flavors of fundamental Dirac fermions (quarks) $q^i$, $i = 1, \ldots, N_f$, has $U(N_f)_{\mathrm{L}} \times U(N_f)_{\mathrm{R}}$ global symmetry. This is easiest to describe if we decompose $q^i$ into left- and right-handed quarks, $q_{\mathrm{L}}^i, q_{\mathrm{R}}^i$, in which case it acts on $q_{\mathrm{L}}^i$ and $q_{\mathrm{R}}^i$ by independent $N_f \times N_f$ unitary matrices and is thus chiral. Two $U(1)$ symmetries are part of this symmetry group: the vector $U(1)$, commonly denoted by $U(1)_{\mathrm{V}}$, which acts on left- and right-handed quarks by the same phase, and the axial $U(1)$, commonly denoted by $U(1)_{\mathrm{A}}$, which acts on left- and right-handed quarks with opposite phases. With the exception of $U(1)_{\mathrm{A}}$, which is broken by an anomaly, the global symmetry group of the Lagrangian of QCD persists in the quantum theory. Starting from the original symmetry group of the classical theory, henceforth denoted by $G_{\mathrm{LRVA}} = SU(N_f)_{\mathrm{L}} \times SU(N_f)_{\mathrm{R}} \times U(1)_{\mathrm{V}} \times U(1)_{\mathrm{A}}$,[1] the remaining symmetry at the quantum level is then $G_{\mathrm{LRV}} = SU(N_f)_{\mathrm{L}} \times SU(N_f)_{\mathrm{R}} \times U(1)_{\mathrm{V}}$.

If we consider nuclear matter at finite temperature, the global symmetry $G_{\mathrm{LRV}}$ (or $G_{\mathrm{LRVA}}$ depending on the fate of $U(1)_{\mathrm{A}}$ at finite temperature) may or may not be broken depending on the temperature. More specifically, chiral symmetry is spontaneously broken at low temperatures due to a non-zero vacuum expectation value for a quark bilinear (quark condensate), namely $\langle \bar{q}_{\mathrm{L}}^i q_{\mathrm{R}}^j \rangle$, which breaks $G_{\mathrm{LRV}}$ (or $G_{\mathrm{LRVA}}$) to $SU(N_f)_{\mathrm{V}} \times U(1)_{\mathrm{V}}$, where $SU(N_f)_{\mathrm{V}}$ is the diagonal subgroup of $G_{\mathrm{LR}} = SU(N_f)_{\mathrm{L}} \times SU(N_f)_{\mathrm{R}}$. At high temperatures, however, the quark condensate is zero and thus $G_{\mathrm{LRV}}$ (or $G_{\mathrm{LRVA}}$) remains unbroken. The order of the associated phase transition at the critical temperature $T_c$, with an order parameter given by the quark condensate, has phenomenological consequences and has been the subject of multiple investigations over the years, starting with the seminal work of Pisarski and Wilczek [1].

Due to its non-chiral nature, the $U(1)_{\mathrm{V}}$ part of $G_{\mathrm{LRV}}$ is expected to play no role in the symmetry breaking and it is common in the literature to neglect it and discuss the group $G_{\mathrm{LR}}$ instead. On the contrary, the $U(1)_{\mathrm{A}}$ part of $G_{\mathrm{LRVA}}$ is of paramount importance. Despite the fact that $U(1)_{\mathrm{A}}$ is anomalous in QCD, it may be effectively restored when non-zero temperature effects are considered [1–4]. If that is the case, then the chiral symmetry to consider in the unbroken phase ($T > T_c$) is $G_{\mathrm{LRA}} = SU(N_f)_{\mathrm{L}} \times SU(N_f)_{\mathrm{R}} \times U(1)_{\mathrm{A}}$ and not $G_{\mathrm{LR}}$. The two cases that have been considered in the literature are those of symmetry $G_{\mathrm{LR}}$ and $G_{\mathrm{LRA}}$.

Quarks are not massless and so consequences obtained in the strict chiral limit are not expected to hold unaltered. However, quark masses much smaller than $T_c$ can be treated as perturbations of the strict chiral case, and then the $N_f = 2$ massless case is of particular interest (with all other quarks treated as infinitely heavy). This is because the critical temperature is given by $T_c \approx 160$ MeV, which is two orders of magnitude larger than the masses of the up and down quarks. $T_c$ is slightly larger than the mass of the strange quark, so one may consider

---

[1]We ignore factors of $\mathbb{Z}_{N_f}$ that result from the isomorphism $U(N) \simeq [SU(N) \times U(1)]/\mathbb{Z}_N$.

the $N_f = 3$ massless case and treat the mass of the strange quark as a perturbation as well, although this approximation may not be as well justified as in the case of up and down quarks.

For the case $N_f = 2$, $G_{LR}$ becomes $SU(2)_L \times SU(2)_R \simeq SO(4)$, while $G_{LRA}$ becomes $SU(2)_L \times SU(2)_R \times U(1)_A \simeq SO(4) \times SO(2)$.[2] In the $G_{LRA}$ case the order of the chiral phase transition was originally suggested to be first order using the $\varepsilon$ expansion [1], and supporting evidence for this conclusion has also been reported [5,6]. However, the conformal bootstrap method applied to this scenario in [7] (for a review see [8]) produced evidence for the existence of a potentially relevant $O(4) \times O(2)$ universality class in $d = 3$ dimensions. The work [9] has also provided evidence in favor of a second order phase transition in the case of effective restoration of $U(1)_A$ using renormalization group methods and resummations. For $N_f = 3$ $G_{LRA}$ is $SU(3)_L \times SU(3)_R \times U(1)_A$ and perturbative methods have not found a fixed point that would open the possibility of a second-order chiral phase transition in three-flavor massless QCD. A recent Monte Carlo analysis for $O(4) \times O(2)$ can be found in [10], whereas the three-flavor case has been studied recently in [11,12]; see also [13] for an overview.

In this work we study conformal field theories (CFTs) with $mn$ complex scalar fields and $U(m) \times U(n)$ global symmetry. The scalar fields are assembled into a complex $m \times n$ matrix $\Phi$ which transforms as a bifundamental under the action of $U(m) \times U(n)$. Our results for the case $m = n = 2, 3$ are of potential relevance to the case of the chiral phase transition of two- and three-flavor massless QCD, respectively.[3] We note that $U(m) \times U(n)$ is the symmetry group that naturally arises in the corresponding Landau–Ginzburg model built with $\Phi$ as a fundamental field [1,5,6,9], which has been discussed in the context of the chiral phase transition of QCD for many years (as discussed above).

Before proceeding to outline our methodology and results, let us point out which results will be relevant if the symmetry of QCD at high temperature is $G_{LRA}$, and which will be relevant if it is $G_{LR}$. Our non-perturbative numerical results, due to the bootstrap, will apply to both cases. The logic is very similar to that of [14]. The relevant difference between the groups $U(n)$ and $SU(n)$ for our purposes is in the existence of the $n$-index Levi–Civita tensor, which is an invariant of $SU(n)$ but not $U(n)$. This will affect the case $U(4) \times U(4)$, which will not be equivalent to $SU(4) \times SU(4)$, in that there will be more sum rules in the $SU(4) \times SU(4)$ case. However, without further assumptions these additional sum rules would not yield different results for the bounds obtained in this work using the $U(4) \times U(4)$ sum rules. On the other hand, our perturbative results will apply to the case of $G_{LRA}$ only. That is because, as discussed in [1] for example, the absence of the $U(1)_A$ symmetry allows one to add the schematic term $g(\det(\Phi) + \det(\Phi^\dagger))$ to the Lagrangian, where $g$ is some coupling. To find controlled perturbative fixed points in our work we necessarily take $g = 0$, which enhances the symmetry to $G_{LRA}$.

We analyze the $U(m) \times U(n)$ model in the standard $\varepsilon$ expansion below $d = 4$ up to three loops and also in the large-$n$ expansion at leading order in $1/n$ using analytic bootstrap methods. Our results include expressions for the scaling dimensions of scalar operators quadratic (bilinear) in $\Phi$ that belong to various irreducible representations (irreps) of the global symmetry group. Dimensions of such non-singlet operators determine crossover exponents. We also use the non-perturbative numerical conformal bootstrap [15] (for a review see [16] and [17]; for a pedagogical introduction see [18]) to obtain upper bounds on various operator dimensions by considering the constraints of unitarity and crossing symmetry in the four-point function of $\Phi$.

For the $N_f = 2$ case our numerical bootstrap bounds coincide with those obtained in [7]. This includes a kink that indicates the possible existence of a CFT with $O(4) \times O(2)$ symmetry (see also [19]). For $N_f = 3$ we also obtain bounds with kinks. Our results provide possible evidence in favor of a second order chiral phase transition in the case of QCD with two or

---

[2]Note that we drop factors of $\mathbb{Z}_2$ in the isomorphism $SU(2) \times SU(2)/\mathbb{Z}_2 \simeq SO(4)$ for convenience of notation.

[3]We may think of $\Phi$ as the order parameter $\langle \bar{q}_L q_R \rangle$ of the phase transition.

three massless flavors. The associated universality classes, however, do not appear to be continuations of the ones predicted by the standard $\varepsilon$ expansion for $U(m) \times U(n)$ theories with $n$ sufficiently larger than $m$.

While not immediately relevant for finite temperature QCD with a small number of massless flavors, we also probe various parameter limits of $U(m) \times U(n)$ symmetric CFTs, such as large $n$ and $m/n$ fixed with both $m$ and $n$ large (for a pedagogical discussion around the significance of fixed points with $m/n$ fixed see [20] and [21]). These are expected to be interesting from the point of view of field theory, given that we make numerous comparisons between perturbative and non-perturbative predictions. Notably, we observe that at large $n$ a lot of our exclusion plots are almost exactly saturated by the perturbative predictions. In some cases this happens even in the absence of any feature in the exclusion plot. Typically, in the numerical conformal bootstrap, kinks are seen as signals of an exclusion bound being saturated by a CFT. In the present work we see explicit examples where this is not strictly necessary. We also see kinks due to theories, that at least naively according to the $\varepsilon$ expansion, should be non-unitary.

This paper is organized as follows. In the next section we review known results regarding theories with $U(m) \times U(n)$ global symmetry in the $\varepsilon = 4 - d$ expansion. In section 3 we work out the group theory required for our analysis of $U(m) \times U(n)$ CFTs. These results are used in section 4 to derive $\varepsilon$ expansion results up to three loops (following the methods developed in [22]), and in section 5 to derive results in the $1/n$ expansion valid in any $d$. In section 6 we obtain non-perturbative numerical bootstrap bounds relevant for $U(m) \times U(n)$ CFTs in $d = 3$. We conclude in section 7. In two appendices we include two different but equivalent ways to derive the crossing equations required in our bootstrap problem, which are of course also equivalent to the way described in section 3 of the main text.

## 2 Fixed points of theories with $U(m) \times U(n)$ global symmetry in the $\varepsilon$ expansion

In the theories we consider, the $mn$ complex scalar fields ($2mn$ real scalar fields) are assembled into an $m \times n$ complex matrix $\Phi_{ar}$, $a = 1 \ldots, m$, $r = 1, \ldots, n$. The Hermitian conjugate of $\Phi_{ar}$ is $\Phi_{ra}^{\dagger}$. Using these fields, we may construct the $U(m) \times U(n)$ invariant Lagrangian [1,5,6]

$$\mathscr{L} = \partial^{\mu}\Phi_{ra}^{\dagger}\,\partial_{\mu}\Phi_{ar} + \tfrac{1}{4}u(\Phi_{ra}^{\dagger}\Phi_{ar})^2 + \tfrac{1}{4}v\,\Phi_{ra}^{\dagger}\Phi_{as}\Phi_{sb}^{\dagger}\Phi_{br}\,, \tag{1}$$

where repeated indices are summed over and we consider up to quartic terms but neglect the mass term $m^2\Phi_{ra}^{\dagger}\Phi_{ar}$.[4] As far as interactions are concerned, we have the two couplings $u$ and $v$. A theory with $v = 0$ preserves $O(2mn)$ symmetry. To examine stability of the quartic potential we choose $m \leqslant n$ without loss of generality and we find that stability requires $u + v \geqslant 0$ if $v < 0$ and $u + \frac{1}{m}v \geqslant 0$ if $v \geqslant 0$.[5]

---

[4]We note that either $U(m)$ or $U(n)$ can realize the $U(1)$ transformation $\Phi_{ar} \to e^{i\alpha}\Phi_{ar}$. Therefore, strictly speaking, the global symmetry group of (1) is $[U(m) \times U(n)]/U(1)$. With this in mind, we will continue to refer to the global symmetry of (1) as $U(m) \times U(n)$ for brevity. For completeness, let us mention that it is also possible to construct a fully $U(m) \times U(n)$ symmetric multiscalar Lagrangian using results of [23]. This would have two distinct mass terms. To our knowledge the existence of a non-trivial fixed point in such a theory in the $\varepsilon$ expansion has not been studied in the literature.

[5]Let $X = \Phi\Phi^{\dagger}$. $X$ is an $m \times m$ Hermitian matrix and by the Cauchy–Schwarz inequality with inner product $\langle A, B \rangle = \text{Tr}(AB)$ for two Hermitian matrices $A, B$, by taking $A = X$ and $B = \mathbb{1}_m$ we find $(\text{Tr}\,X)^2 \leqslant m\,\text{Tr}\,X^2$. Additionally, since $X$ is positive-definite, we have $(\text{Tr}\,X)^2 \geqslant \text{Tr}\,X^2$.

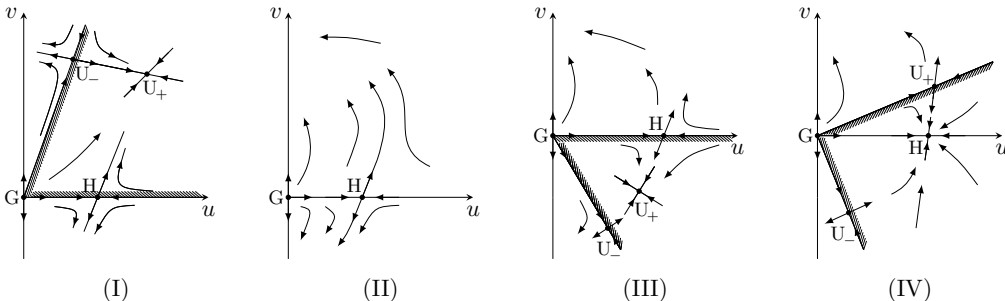

Figure 1: Schematic flow diagrams corresponding to the four regimes mentioned in the text. The location of the fixed points should not be viewed as precise, but the fixed points are placed at locations consistent with the sign of the coupling $v$ for which they occur. The region between the hatched lines in each diagram represents the basin of attraction of the stable fixed point.

The number of real fixed points of the Lagrangian (1) depends on the values of $m$ and $n$. There are four regimes:

(I) For $n > n^+(m)$ there are four real fixed points (Gaussian, $O(2mn)$, $U_-$, $U_+$). Stable[6] fixed point: $U_+$.

(II) For $n^-(m) < n < n^+(m)$ there are two real fixed points (Gaussian and $O(2mn)$). They are both unstable.

(III) For $n_H(m) < n < n^-(m)$ there are four real fixed points (Gaussian, $O(2mn)$, $U_-$, $U_+$). Stable fixed point: $U_+$.

(IV) For $n < n_H(m)$ there are four real fixed points (Gaussian, $O(2mn)$, $U_-$, $U_+$). Stable fixed point: $O(2mn)$.

The Gaussian fixed point has $u = v = 0$, while the $O(2mn)$ fixed point has $u > 0, v = 0$. The fully-interacting fixed points (i.e. the ones besides Gaussian and $O(2mn)$) both have $uv \neq 0$ and $U(m) \times U(n)$ global symmetry. These fixed points move around in the $u$-$v$ coupling plane as $m, n$ change. For every $m$ there is a value of $n$, indicated by $n^+(m)$ above, for which $U_-$ and $U_+$ collide in the real $u$-$v$ plane and subsequently move to the complex $u$-$v$ plane as we go below $n^+(m)$. For $n > n^+(m)$ the $U_+$ fixed point is stable and has $v > 0$, as does $U_-$, but for $n < n^+(m)$ there is no real stable fixed point. However, for some $n^-(m) < n^+(m)$ these two fixed points reappear in the $u$-$v$ plane—this time they have $v < 0$ and $U_+$ is again stable. Furthermore, there is a value $n^H(m) < n^-(m)$ below which the $O(2mn)$ fixed point is stable, since one of the fully interacting fixed points of the $n^H(m) < n < n^-(m)$ regime crosses the $v = 0$ line and acquires $v > 0$, while the other remains with $v < 0$. These four regimes are depicted in Fig. 1.

The values of $n^\pm(m)$ can be estimated in the $\varepsilon$ expansion:

$$n^\pm(m) = 5m \pm 2\sqrt{6(m-1)(m+1)} - \left(5m \pm \frac{(5m-4)(5m+4)}{2\sqrt{6(m-1)(m+1)}}\right)\varepsilon + O(\varepsilon^2). \qquad (2)$$

In a recent paper, these results have been extended to six loops, or order $\varepsilon^5$ [6]. The value of $n^+(m)$ is of interest due to applications to $m$-flavor QCD. In particular, if $n^+(m) < m$ in $d = 3$, then there exists a unitary 3D CFT with $U(m) \times U(m)$ global symmetry (corresponding to $U_+$

---

[6]A fixed point with only one relevant scalar singlet operator, namely the mass operator $\Phi^\dagger_{ra}\Phi_{ar}$, is called stable.

in regime (I) above). However, the essential conclusion of the $\varepsilon$ expansion after resummations is that $n^+(m) > m$ [5,6] in $d = 3$. This has been used to argue that there is no stable unitary fixed point of (1) for $m = n$ that could describe the $m$-flavor chiral phase transition of QCD, which requires a stable fixed point as well as $v > 0$ for the appropriate symmetry-breaking pattern and thus could not lie in regimes (II), (III) or (IV).

## 3 Group theoretic considerations

In this section we describe the group theoretic ingredients that will allow us to derive results in the $\varepsilon$ expansion as well as the crossing equation that we will use for our analytic and numerical bootstrap studies.

For our purposes[7] it is more economical (index-wise) to consider the replacement [24]

$$\Phi_{ar} = T_{ar}{}^i \phi_i \,, \qquad \Phi^\dagger_{ra} = T^\dagger_{ra}{}^i \phi_i \,, \qquad i = 1, \ldots, 2mn \,, \tag{3}$$

where the $2mn$ complex $m \times n$ matrices $T_{ar}{}^i$ encode the complex nature of $\Phi_{ar}$, while the $2mn$ fields $\phi_i$ are real. Essentially, the fields $\Phi$ and $\Phi^\dagger$ are repackaged into their real and complex parts, schematically $(\Phi + \Phi^\dagger)$ and $-i(\Phi - \Phi^\dagger)$.[8] The $T$ matrices satisfy

$$T^\dagger_{ra}{}^i T_{bs}{}^i = \delta_{ab}\delta_{rs} \,, \qquad T_{ar}{}^i T_{bs}{}^i = 0 \,, \qquad T^\dagger_{ra}{}^i T^\dagger_{sb}{}^i = 0 \,,$$
$$T^\dagger_{ra}{}^i T_{ar}{}^j + T^\dagger_{ra}{}^j T_{ar}{}^i = \mathrm{Tr}(T^{\dagger i} T^j + T^{\dagger j} T^i) = \delta^{ij} \,. \tag{4}$$

### 3.1 Rank-four invariant tensors

The $T$ matrices allow us to construct the rank-four (in the indices $i, j, \ldots$) invariant tensors of $U(m) \times U(n)$. These can be used to derive a variety of results in the $\varepsilon$ expansion up to three loops as described in [22].[9]

For $m \neq n$ there is one fully symmetric traceless rank-four primitive invariant tensor, $\zeta_{ijkl}$, three rank-four primitive invariant tensors $\omega_{u,ijkl}$, $u = 1, 2, 3$, with symmetry properties

$$\omega_{u,ijkl} = \omega_{u,jikl} \,, \qquad \omega_{u,klij} = \omega_{u,ijkl} \,, \qquad \omega_{u,i(jkl)} = 0 \,, \qquad \omega_{u,iikl} = 0 \,, \tag{5}$$

and two fully antisymmetric rank-four primitive invariant tensors, $\psi_{x,ijkl}$, $x = 1, 2$. With the addition of the non-primitive rank-four invariant tensors $\delta_{ij}\delta_{kl}$, $\delta_{ik}\delta_{jl}$ and $\delta_{il}\delta_{jk}$ we have a total of 12 independent rank-four invariant tensors.[10] One can check that products of these 12 tensors with four free indices (such as e.g. $\omega_{u,ijkl}\zeta_{klmn}$) close on themselves, i.e. they do not produce any additional tensors. For $m = n$ we can treat the two $U(n)$'s as distinguishable or indistinguishable. In the latter case one of the $\omega$ and one of the $\psi$ tensors disappear and we are left with 9 independent invariant tensors.

With the use of the real scalar fields $\phi_i$, the $U(m) \times U(n)$ invariant Lagrangian takes the form[11]

$$\mathcal{L} = \tfrac{1}{2}\partial^\mu \phi_i \partial_\mu \phi_i + \tfrac{1}{8}\lambda(\phi^2)^2 + \tfrac{1}{24}g\,\zeta_{ijkl}\phi_i\phi_j\phi_k\phi_l \,, \tag{6}$$

---

[7]Converting to a one-index real-field notation allows us to readily extract results from existing $\varepsilon$ expansion computations in the literature. The reader directly interested in numerical bootstrap sum rules may see Appendix A.

[8]As an explicit example, consider $\Phi_{11} = \phi_1 + i\phi_2$; then $T_{11}{}^1 = 1$ and $T_{11}{}^2 = i$. Also, all other elements, such as e.g. $T_{12}{}^1$ or $T_{44}{}^1$, are zero. With this example in mind, the reader may convince themselves by inspection that (4) holds (up to normalization).

[9]With the recent work of [24] some of these results can be extended to six loops.

[10]There are two inequivalent index permutations and therefore two independent invariant tensors for each $\omega$.

[11]We consider terms up to quartic in $\phi$ and neglect the mass term $\tfrac{1}{2}m^2\phi^2$.

where $\phi^2 = \phi_i \phi_i$ and

$$\zeta_{ijkl} = \tfrac{1}{2} \operatorname{Tr}\left[ T^\dagger_{(i} T_j T^\dagger_k T_{l)} \right] - \frac{m+n}{mn+1} (\delta_{ij}\delta_{kl} + \delta_{ik}\delta_{jl} + \delta_{il}\delta_{jk}), \tag{7}$$

where parentheses around indices are used to denote symmetrization of the enclosed indices.[12] Due to cyclicity of the trace there are 12 distinct terms among the 24 produced by the symmetrization of the indices. Choosing, without loss of generality, $m \leqslant n$, the bound

$$\frac{3(m-1)(m+1)}{m(mn+1)}(\phi^2)^2 \leqslant \zeta_{ijkl}\phi_i\phi_j\phi_k\phi_l \leqslant \frac{3(m-1)(n-1)}{mn+1}(\phi^2)^2 \tag{8}$$

is satisfied. The couplings $\lambda, g$ of (6) are related to the couplings $u, v$ of (1) via

$$\lambda = \tfrac{1}{2}\Big(u + \frac{m+n}{mn+1}v\Big), \qquad g = \tfrac{1}{2}v. \tag{9}$$

The tensor $\zeta_{ijkl}$ satisfies

$$\zeta_{ijmn}\zeta_{mnkl} = \frac{1}{2mn-1}a\big(mn(\delta_{ik}\delta_{jl} + \delta_{il}\delta_{jk}) - \delta_{ij}\delta_{kl}\big) + e^u\,\omega_{u,ijkl} + b\,\zeta_{ijkl}, \tag{10}$$

with

$$a = \frac{6(m-1)(m+1)(n-1)(n+1)}{(mn+1)^2},$$
$$e^1 = \frac{2(m+n)}{3}, \qquad e^2 = m-n, \qquad e^3 = \frac{2}{3}(2m+2n+3), \tag{11}$$
$$b = \frac{2(m+n)(mn-5)}{3(mn+1)},$$

where

$$\begin{aligned}
\omega_{1,ijkl} = {}& \operatorname*{Sym}_{ij}\operatorname*{Sym}_{kl}\Big\{ \tfrac{1}{2}\operatorname{Tr}\big(T^\dagger_i T_j T^\dagger_k T_l - T^\dagger_i T_k T^\dagger_j T_l - T^\dagger_k T_i T^\dagger_l T_j + T^\dagger_i T_l T^\dagger_k T_j\big) \\
& - 2\big[\operatorname{Tr}(T^\dagger_i T_k)\operatorname{Tr}(T^\dagger_j T_l) + \operatorname{Tr}(T^\dagger_k T_i)\operatorname{Tr}(T^\dagger_l T_j) - 2\operatorname{Tr}(T^\dagger_i T_j)\operatorname{Tr}(T^\dagger_k T_l)\big]\Big\} \\
& + \frac{4mn+m+n}{2(2mn-1)}(\delta_{ik}\delta_{jl} + \delta_{il}\delta_{jk} - 2\delta_{ij}\delta_{kl}), \\
\omega_{2,ijkl} = {}& \operatorname*{Sym}_{ij}\operatorname*{Sym}_{kl}\big[\operatorname{Tr}(T^\dagger_i T_j T^\dagger_k T_l - T^\dagger_i T_l T^\dagger_k T_j)\big] + \frac{m-n}{2mn-1}(\delta_{ik}\delta_{jl} + \delta_{il}\delta_{jk} - 2\delta_{ij}\delta_{kl}), \\
\omega_{3,ijkl} = {}& \operatorname*{Sym}_{ij}\operatorname*{Sym}_{kl}\big[\operatorname{Tr}(T^\dagger_i T_k)\operatorname{Tr}(T^\dagger_j T_l) + \operatorname{Tr}(T^\dagger_k T_i)\operatorname{Tr}(T^\dagger_l T_j) - 2\operatorname{Tr}(T^\dagger_i T_j)\operatorname{Tr}(T^\dagger_k T_l)\big] \\
& - \frac{mn}{2mn-1}(\delta_{ik}\delta_{jl} + \delta_{il}\delta_{jk} - 2\delta_{ij}\delta_{kl}).
\end{aligned} \tag{12}$$

The operator $\operatorname{Sym}_{ij}$ simply symmetrizes the indices $i, j$ of the expression on which it acts.

There are further identities like (10) involving the $\omega_u$ tensors in the left-hand side. These take the form

$$\omega_{u,ijmn}\zeta_{mnkl} = f_u{}^v\,\omega_{v,ijkl} + h_u\,\zeta_{ijkl}, \tag{13}$$

---

[12]Here and hereafter symmetrization and antisymmetrization of indices is defined without an overall factorial normalization factor.

with

$$f_1{}^1 = \frac{m^2 n + mn^2 + 8mn - 5m - 5n + 8}{3(mn + 1)}, \qquad f_1{}^2 = \tfrac{1}{2}(m - n), \qquad f_1{}^3 = \tfrac{2}{3}(m + n + 5),$$

$$f_2{}^1 = \tfrac{2}{3}(m - n), \qquad f_2{}^2 = \frac{(m + n)(mn - 1)}{mn + 1}, \qquad f_2{}^3 = \tfrac{4}{3}(m - n),$$

$$f_3{}^1 = -\tfrac{4}{3}, \qquad f_3{}^2 = 0, \qquad f_3{}^3 = -\frac{2(4mn + 3m + 3n + 4)}{3(mn + 1)}, \tag{14}$$

$$h_1 = \frac{2(m + n + 2)(mn + 1)}{3(2mn - 1)}, \qquad h_2 = \frac{4(m - n)(mn + 1)}{3(2mn - 1)}, \qquad h_3 = -\frac{2(mn + 1)}{3(2mn - 1)},$$

and

$$\omega_{u,ijmn}\, \omega_{v,mnkl} = \frac{1}{2mn - 1} a'_{uv}\big(mn(\delta_{ik}\delta_{jl} + \delta_{il}\delta_{jk}) - \delta_{ij}\delta_{kl}\big) + e'_{uv}{}^w \omega_{w,ijkl} + b'_{uv}\, \zeta_{ijkl}, \tag{15}$$

with the parameters appearing also determined but not quoted here.

Further relations involving the $\omega_u$ tensors in the left-hand side require the $\psi_x$ tensors in the right-hand side:

$$\omega_{u,imjn}\, \omega_{v,kmln} = \frac{1}{(mn + 1)(2mn - 1)^2}\big(\tilde{a}_{uv}\, \delta_{ik}\delta_{jl} + \hat{a}_{uv}\, \delta_{il}\delta_{jk} + \breve{a}_{uv}\, \delta_{ij}\delta_{kl}\big)$$
$$+ \frac{1}{2mn - 1}(\tilde{e}_{uv}{}^w \omega_{w,ijkl} + \hat{e}_{uv}{}^w \omega_{w,ikjl}) + b''_{uv}\, \zeta_{ijkl} + d_{uv}{}^x \psi_{x,ijkl}, \tag{16}$$

and there are similar relations for $\omega_{u,imjn}\psi_{x,mnkl}$ and $\psi_{x,ijmn}\psi_{y,mnkl}$. The tensors $\psi_x$ are given by

$$\psi_{1,ijkl} = \mathrm{Tr}\big(T^\dagger_{[i}T_j T^\dagger_k T_{l]}\big), \qquad \psi_{2,ijkl} = \mathrm{Tr}\big(T^\dagger_{[i}T_j\big)\mathrm{Tr}\big(T^\dagger_k T_{l]}\big),$$

where brackets around indices are used to denote antisymmetrization of the enclosed indices. There are 12 distinct terms in $\psi_1$ and 6 in $\psi_2$.

Finally, there is a relevant identity involving four $\zeta$ tensors,

$$\zeta_{ii'j'k'}\, \zeta_{ji'l'm'}\, \zeta_{kj'l'n'}\, \zeta_{lk'm'n'} = \tfrac{1}{4}\mathcal{A}\big(\delta_{ij}\delta_{kl} + \delta_{ik}\delta_{jl} + \delta_{il}\delta_{jk}\big) + c\, \zeta_{ijkl}, \tag{17}$$

with

$$\mathcal{A} = \frac{1}{N - 1} a\big((N - 2)a + 2(N - 1)b^2\big) - a\, e^u h_u,$$
$$c = -\frac{2(m + n)(5m^3 n^3 - m^2 n^2 + 71mn - 20m^3 n - 20mn^3 + 4m^2 + 4n^2 + 29)}{(mn + 1)^3}. \tag{18}$$

When $m = n$ the two $U(n)$ factors in the global symmetry group $U(n) \times U(n)$ may be treated as distinguishable or indistinguishable. In the former case one simply needs to take $m = n$ in the various expressions given in this work. In the latter case the indices carried by $\Phi$ are indistinguishable, the global symmetry is enhanced to $U(n)^2 \rtimes \mathbb{Z}_2$ and the tensors $\omega_2$ and $\psi_1$ vanish. We will comment on this case separately at various points below.

## 3.2 Rank-four projectors

To derive the projectors we will convert to invariant tensors that are not traceless. This is not essential and is only done for simplicity of the expressions for the projectors. We thus define

$$\tilde{\zeta}_{ijkl} = \zeta_{ijkl} + \frac{m+n}{mn+1}(\delta_{ik}\delta_{jl} + \delta_{il}\delta_{jk} + \delta_{ij}\delta_{kl}),$$

$$\tilde{\omega}_{1,ijkl} = \omega_{1,ijkl} - \frac{m+n+2}{2(2mn-1)}(\delta_{ik}\delta_{jl} + \delta_{il}\delta_{jk} - 2\delta_{ij}\delta_{kl}),$$

$$\tilde{\omega}_{2,ijkl} = \omega_{2,ijkl} - \frac{m-n}{2mn-1}(\delta_{ik}\delta_{jl} + \delta_{il}\delta_{jk} - 2\delta_{ij}\delta_{kl}),$$

$$\tilde{\omega}_{3,ijkl} = \omega_{3,ijkl} + \frac{1}{2(2mn-1)}(\delta_{ik}\delta_{jl} + \delta_{il}\delta_{jk} - 2\delta_{ij}\delta_{kl}).$$

$$(19)$$

The $\tilde{\omega}_u$ tensors satisfy all but the last of the properties in (5). Using these tensors one can define the 12 rank-four projectors

$$P_{ijkl}^{S_{\text{even}}} = \frac{1}{2mn}\delta_{ij}\delta_{kl},$$

$$P_{ijkl}^{S_{\text{odd}}} = \frac{1}{3mn}\left(\tilde{\omega}_{3,ijkl} + 2\tilde{\omega}_{3,ikjl} + \psi_{2,ijkl}\right),$$

$$P_{ijkl}^{RS_{\text{even}}} = -\frac{1}{2mn}\delta_{ij}\delta_{kl} + \frac{1}{6n}(\tilde{\zeta}_{ijkl} + \tilde{\omega}_{1,ijkl} - \tfrac{3}{2}\tilde{\omega}_{2,ijkl} + 2\tilde{\omega}_{3,ijkl}),$$

$$P_{ijkl}^{RS_{\text{odd}}} = -\frac{1}{6n}(\tilde{\omega}_{1,ijkl} + 2\tilde{\omega}_{1,ikjl} + \tfrac{1}{2}\tilde{\omega}_{2,ijkl} + \tilde{\omega}_{2,ikjl} + \psi_{1,ijkl})$$
$$\qquad - \frac{1}{3mn}[(m+1)(\tilde{\omega}_{3,ijkl} + 2\tilde{\omega}_{3,ikjl}) + \psi_{2,ijkl}],$$

$$P_{ijkl}^{SR_{\text{even}}} = -\frac{1}{2mn}\delta_{ij}\delta_{kl} + \frac{1}{6m}(\tilde{\zeta}_{ijkl} + \tilde{\omega}_{1,ijkl} + \tfrac{3}{2}\tilde{\omega}_{2,ijkl} + 2\tilde{\omega}_{3,ijkl}),$$

$$P_{ijkl}^{SR_{\text{odd}}} = -\frac{1}{6m}(\tilde{\omega}_{1,ijkl} + 2\tilde{\omega}_{1,ikjl} - \tfrac{1}{2}\tilde{\omega}_{2,ijkl} - \tilde{\omega}_{2,ikjl} + \psi_{1,ijkl})$$
$$\qquad - \frac{1}{3mn}[(n+1)(\tilde{\omega}_{3,ijkl} + 2\tilde{\omega}_{3,ikjl}) + \psi_{2,ijkl}],$$

$$P_{ijkl}^{RR_{\text{even}}} = \tfrac{1}{4}(\delta_{ik}\delta_{jl} + \delta_{il}\delta_{jk}) - \frac{1}{2mn}\delta_{ij}\delta_{kl} - \frac{m+n}{6mn}\tilde{\zeta}_{ijkl}$$
$$\qquad - \frac{1}{6mn}[(m+n)\tilde{\omega}_{1,ijkl} - \tfrac{3}{2}(m-n)\tilde{\omega}_{2,ijkl} + (3mn+2m+2n)\tilde{\omega}_{3,ijkl}],$$

$$P_{ijkl}^{RR_{\text{odd}}} = -\tfrac{1}{4}(\delta_{ik}\delta_{jl} - \delta_{il}\delta_{jk}) + \frac{1}{6mn}[(m+n)(\tilde{\omega}_{1,ijkl} + 2\tilde{\omega}_{1,ikjl}) + \tfrac{1}{2}(m-n)(\tilde{\omega}_{2,ijkl} + 2\tilde{\omega}_{2,ikjl})$$
$$\qquad + (mn+2m+2n+2)(\tilde{\omega}_{3,ijkl} + 2\tilde{\omega}_{3,ikjl})] + \frac{m-n}{6mn}\psi_{1,ijkl} - \frac{mn-1}{3mn}\psi_{2,ijkl},$$

$$P_{ijkl}^{TT_{\text{even}}} = \tfrac{1}{8}(\delta_{ik}\delta_{jl} + \delta_{il}\delta_{jk}) + \tfrac{1}{12}\tilde{\zeta}_{ijkl} - \tfrac{1}{6}\tilde{\omega}_{1,ijkl} - \tfrac{1}{12}\tilde{\omega}_{3,ijkl},$$

$$P_{ijkl}^{TA_{\text{odd}}} = -\tfrac{1}{8}(\delta_{ik}\delta_{jl} - \delta_{il}\delta_{jk}) + \tfrac{1}{12}(\tilde{\omega}_{2,ijkl} + 2\tilde{\omega}_{2,ikjl})$$
$$\qquad - \tfrac{1}{12}(\tilde{\omega}_{3,ijkl} + 2\tilde{\omega}_{3,ikjl}) - \tfrac{1}{12}\psi_{1,ijkl} + \tfrac{1}{6}\psi_{2,ijkl},$$

$$P_{ijkl}^{AT_{\text{odd}}} = -\tfrac{1}{8}(\delta_{ik}\delta_{jl} - \delta_{il}\delta_{jk})$$
$$\qquad - \tfrac{1}{12}(\tilde{\omega}_{2,ijkl} + 2\tilde{\omega}_{2,ikjl}) - \tfrac{1}{12}(\tilde{\omega}_{3,ijkl} + 2\tilde{\omega}_{3,ikjl}) + \tfrac{1}{12}\psi_{1,ijkl} + \tfrac{1}{6}\psi_{2,ijkl},$$

$$P_{ijkl}^{AA_{\text{even}}} = \tfrac{1}{8}(\delta_{ik}\delta_{jl} + \delta_{il}\delta_{jk}) - \tfrac{1}{12}\tilde{\zeta}_{ijkl} + \tfrac{1}{6}\tilde{\omega}_{1,ijkl} + \tfrac{7}{12}\tilde{\omega}_{3,ijkl},$$

$$(20)$$

where the subscripts "even" and "odd" refer to the Lorentz spins with which the corresponding irreps appear in the $\phi_i \times \phi_j$ OPE. Note that $\tilde{\omega}_{u,jikl} + 2\tilde{\omega}_{u,jkil} = -\tilde{\omega}_{u,ijkl} - 2\tilde{\omega}_{u,ikjl}$ so that the

"odd" projectors are odd under $i \leftrightarrow j$ (and under $k \leftrightarrow l$). The projectors satisfy

$$P_{ijmn}^I P_{nmkl}^J = P_{ijkl}^I \delta^{IJ}, \qquad \sum_I P_{ijkl}^I = \delta_{il}\delta_{jk}, \qquad P_{ijkl}^I \delta_{il}\delta_{jk} = d_r^I, \tag{21}$$

where $d_r^I$ is the dimension of the irrep indexed by $I$:

$$\begin{aligned}
\vec{d}_r = \Big( &[1]_2, [(m-1)(m+1)]_2, [(n-1)(n+1)]_2, [(m-1)(m+1)(n-1)(n+1)]_2, \\
&\tfrac{1}{2}m(m+1)n(n+1), \tfrac{1}{2}m(m+1)(n-1)n, \tfrac{1}{2}(m-1)mn(n+1), \tfrac{1}{2}(m-1)m(n-1)n \Big),
\end{aligned} \tag{22}$$

where by $[x]_2$ we mean that $x$ appears two consecutive times.

In the case where $m = n$ and we treat the two $U(n)$ factors as indistinguishable, then, as a consequence of the disappearance of $\tilde{\omega}_2$ and $\psi_1$, instead of separate projectors $P^{RS_\text{even}}$ and $P^{SR_\text{even}}$ we only have the sum $P^{RS_\text{even}} + P^{SR_\text{even}}$,[13] and the same happens for $P^{RS_\text{odd}}, P^{SR_\text{odd}}$ and $P^{TA_\text{odd}}, P^{AT_\text{odd}}$. Consequently, we have a total of 9 independent rank-four projectors.

In Appendices A and B we give projectors using different ways of parametrizing the scalar fields. Parametrizing the field with a suitable number of indices, one is able to express the projectors solely in terms of Kronecker deltas for both real and complex fields. These have the advantage of being the most straightforward tensors one can write down.

## 4 Results in the $\varepsilon$ expansion

The results of the previous section suffice to determine beta functions and anomalous dimensions up to three loops following [22]. With the rescalings $(\lambda, g) \to 16\pi^2(\lambda, g)$ and using $N = 2mn$, these are

$$\begin{aligned}
\beta_\lambda^{(1)} &= -\varepsilon\,\lambda + (N+8)\lambda^2 + a\,g^2, \\
\beta_g^{(1)} &= -\varepsilon\,g + 12\,\lambda\,g + 3\,b\,g^2,
\end{aligned} \tag{23}$$

$$\begin{aligned}
\beta_\lambda^{(2)} &= -3(3N+14)\lambda^3 - \tfrac{1}{6}(5N+82)a\,\lambda g^2 - 2\,a b\,g^3, \\
\beta_g^{(2)} &= -(5N+82)\lambda^2 g + \tfrac{1}{6(N-1)}(N^2 - 17N + 34)a\,g^3 - 6\,b(6\lambda g^2 + b\,g^3) + 3\,e^u h_u\,g^3,
\end{aligned} \tag{24}$$

and

$$\begin{aligned}
\beta_\lambda^{(3)} &= \tfrac{1}{8}(33N^2 + 922N + 2960)\lambda^4 + 12(5N+22)\zeta_3\lambda^4 \\
&\quad + \tfrac{1}{16}(N^2 + 500N + 3492)a\,\lambda^2 g^2 + 12(N+14)\zeta_3\,a\,\lambda^2 g^2 \\
&\quad + \tfrac{1}{8}(27N + 470)a\,b\,\lambda g^3 + 48\zeta_3\,a\,b\,\lambda g^3 \\
&\quad - \tfrac{1}{16(N-1)}(7N^2 - 33N + 114)a^2\,g^4 + \tfrac{13}{2}\,a\,b^2\,g^4 - \tfrac{11}{2}\,a\,e^u h_u\,g^4 + 3\,\mathcal{A}\,\zeta_3\,g^4, \\
\beta_g^{(3)} &= -\tfrac{1}{4}(13N^2 - 368N - 3284)\lambda^3 g + 48(N+14)\zeta_3\,\lambda^3 g \\
&\quad + \tfrac{3}{8}(43N + 1334)b\,\lambda^2 g^2 + 432\zeta_3\,b\,\lambda^2 g^2 \\
&\quad + \tfrac{3}{N-1}\big(3N^2 + 33N - 50\big)a\,\lambda g^3 + 156\,b^2\,\lambda g^3 - 42\,e^u h_u\,\lambda g^3 + 72\,\mathcal{A}/a\,\zeta_3\,\lambda g^3 \\
&\quad - \tfrac{1}{16(N-1)}(11N^2 - 289N + 626)a\,b\,g^4 + \tfrac{39}{2}\,b^3\,g^4 - \tfrac{3}{4}\big(29\,b\,e^u h_u - 8\,e^u f_u^v h_v\big)g^4 \\
&\quad + 12\zeta_3\,c\,g^4.
\end{aligned} \tag{25}$$

The $\phi$ anomalous dimension matrix is $\gamma_\phi \mathbb{1}_N$ with

$$\gamma_\phi^{(2)} = \tfrac{1}{4}(N+2)\big(\lambda^2 + \tfrac{1}{6}\,a\,g^2\big), \tag{26}$$

---

[13]in which case we will refer to the corresponding irrep as *RSSR*.

and
$$\gamma_\phi^{(3)} = -\tfrac{1}{16}(N+2)(N+8)\lambda^3 - \tfrac{1}{32}(N+2)(6a\,\lambda g^2 + a\,b\,g^3). \tag{27}$$

Relevant operators quadratic in $\phi$ can be considered by extending (6) by
$$\mathscr{L} \to \mathscr{L} + \tfrac{1}{2}\sigma\,\phi^2 + \tfrac{1}{2}\rho_{ij}\,\phi_i\phi_j, \qquad \rho_{ii} = 0. \tag{28}$$

The corresponding beta functions for $\sigma, \rho$ are then [22]

$$\beta_\sigma^{(1)} = (N+2)\lambda\,\sigma, \qquad \beta_\sigma^{(2)} = -\tfrac{5}{2}(N+2)(\lambda^2 + \tfrac{1}{6}a\,g^2)\sigma,$$
$$\beta_\sigma^{(3)} = \tfrac{1}{16}(N+2)\big(12(5N+37)\lambda^3 + (N+164)a\,g^2\lambda + 27\,ab\,g^3\big)\sigma,$$
$$\beta_{\rho,ij}^{(1)} = 2\lambda\,\rho_{ij} + g\,\zeta_{ijkl}\,\rho_{kl},$$
$$\beta_{\rho,ij}^{(2)} = -\tfrac{1}{2}\big((N+10)\lambda^2 - \tfrac{N^2-5N+10}{6(N-1)}a\,g^2\big)\rho_{ij} - (4\lambda g + b\,g^2)\zeta_{ijkl}\,\rho_{kl} + \tfrac{1}{2}e^u\,\omega_{u,ijkl}\,\rho_{kl},$$

$$\begin{aligned}
\beta_{\rho,ij}^{(3)} = &-\tfrac{1}{4}\big(\tfrac{1}{2}(5N^2-84N-444)\lambda^3 - \tfrac{5N^2+65N-82}{N-1}a\,g^2\lambda + \tfrac{N^2-35N+54}{4(N-1)}ab\,g^3\big)\rho_{ij} \\
&+\tfrac{1}{8}\big(3(9N+146)g\,\lambda^2 + 192\,b\,g^2\lambda - \tfrac{3N^2-25N+66}{2(N-1)}a\,g^3 + 32\,b^2\,g^3 - 22\,e^u h_u\,g^3\big)\zeta_{ijkl}\,\rho_{kl} \\
&-\big(3\,g^2\lambda + \tfrac{5}{4}b\,g^3\big)e^u\,\omega_{u,ijkl}\,\rho_{kl} + g^3\,e^u f_u^{\ v}\,\omega_{v,ijkl}\,\rho_{kl}.
\end{aligned} \tag{29}$$

In general, $\beta_\sigma = \gamma_\sigma\sigma$, but anomalous dimensions for $\rho$ are determined by the eigenvalue problem
$$\zeta_{ijkl}\,v_{kl} = \mu\,v_{ij}, \qquad \omega_{u,ijkl}\,v_{kl} = \mu_u\,v_{ij}, \qquad v_{ij} = v_{ji}, \ v_{ii} = 0, \tag{30}$$
requiring, using (10), (13) and (15),
$$\mu^2 = e^u\mu_u + b\,\mu + \tfrac{N}{N-1}a, \quad \mu\,\mu_u = f_u^{\ v}\mu_v + h_u\,\mu, \quad \mu_u\,\mu_v = e'^{\ w}_{uv}\mu_w + b'_{uv}\,\mu + \tfrac{N}{N-1}a'_{uv}. \tag{31}$$

The results (23)–(27) and (29), (31), for the appropriate $a, b, c, e^u, f_u^{\ v}, h^u, a'_{uv}, b'_{uv}$ and $e'^{\ w}_{uv}$, apply to any scalar theory with a global symmetry group that has a unique rank-four symmetric traceless primitive invariant tensor [22]. In this work we will focus on the two fixed points of (6), labeled $U_\pm$, that preserve $U(m) \times U(n)$ symmetry.[14] At leading order in the $\varepsilon$ expansion,

$$\begin{aligned}
\lambda_\pm^{(1)} &= \frac{1}{4(mn+1)D_{mn}}\Big(A_{mn} \pm B_{mn}\sqrt{R_{mn}}\Big)\varepsilon, \\
g_\pm^{(1)} &= \frac{1}{2D_{mn}}\Big(B_{mn} \mp 3\sqrt{R_{mn}}\Big)\varepsilon,
\end{aligned} \tag{32}$$

where
$$A_{mn} = 2m^2n^2 + 14mn + m^3n + mn^3 - 11m^2 - 11n^2 + 36, \qquad B_{mn} = m^2n + mn^2 - 5m - 5n,$$
$$D_{mn} = 2m^2n^2 - 16mn + m^3n + mn^3 - 8m^2 - 8n^2 + 108, \qquad R_{mn} = m^2 + n^2 - 10mn + 24. \tag{33}$$

The two fixed points coincide when $R_{mn} = 0$, in which case the upper bound of [25] on the quantity $\lambda_{ijkl}\lambda_{ijkl}$ at leading order in the $\varepsilon$ expansion, namely $\lambda_{ijkl}\lambda_{ijkl} \leqslant \tfrac{1}{8}N\varepsilon^2$, is saturated. Using [26] we find that the Diophantine equation $R_{mn} = 0$ has an infinite number of positive integer solutions given by (without loss of generality we assume $m < n$)

$$\begin{aligned}
m_{i+1} &= n_i, \quad n_{i+1} = -m_i + 10n_i, \quad i = 1, 2, \ldots, \\
m_1 &= 1, \quad n_1 = 5.
\end{aligned} \tag{34}$$

---

[14]There are two further fixed points of (6): the free theory and the $O(2mn)$ model.

The solution with smallest $N$ is $m = 5$, $n = 49$, $N = 490$, since for $m = 1$ $g_{\pm}^{(1)}$ is singular at $n = 5$. When the $U_{\pm}$ fixed points coincide they annihilate and move off to the complex $(\lambda, g)$ plane as discussed in section 2. The solutions (34) correspond to $n^+$ of (2) at $\varepsilon = 0$.

To present results in compact form we will assume, without loss of generality, that $m < n$ and present anomalous dimensions in a large-$n$ expansion up to three loops but at leading order in $1/n$. The full unexpanded in $n$ results are straightforward to compute with our methods and they are included in an ancillary file. The anomalous dimension of $\phi$ is

$$\gamma_{\phi,+} = \frac{m}{8n}\left(\varepsilon^2 - \tfrac{1}{4}\varepsilon^3\right), \qquad \gamma_{\phi,-} = \frac{(m-1)(m+1)}{8mn}\left(\varepsilon^2 - \tfrac{1}{4}\varepsilon^3\right). \tag{35}$$

The dimension of $\phi$ at the fixed points $U_{\pm}$ is equal to $\Delta_{\pm} = 1 - \tfrac{1}{2}\varepsilon + \gamma_{\pm}$.

For the $\phi^2$ operator (leading scalar in the irrep $S_{\text{even}}$ above) we find

$$\gamma_{\sigma,+} = \varepsilon - \frac{m}{n}\left(3\varepsilon - \tfrac{13}{4}\varepsilon^2 + \tfrac{3}{16}\varepsilon^3\right), \qquad \gamma_{\sigma,-} = \frac{(m-1)(m+1)}{mn}\left(3\varepsilon - \tfrac{13}{4}\varepsilon^2 + \tfrac{3}{16}\varepsilon^3\right). \tag{36}$$

Finally, for the $\rho_{ij}\phi_i\phi_j$ operators we find a decomposition into five distinct cases, with

$$\gamma_{\rho_1,+} = \varepsilon - \frac{m}{n}\left(\varepsilon - \tfrac{5}{4}\varepsilon^2 + \tfrac{3}{16}\varepsilon^3\right), \quad \gamma_{\rho_1,-} = \varepsilon - \frac{1}{mn}\left[(m^2 - 5)\varepsilon - \tfrac{5}{4}(m^2 - 5)\varepsilon^2 + \tfrac{1}{16}(3m^2 - 11)\varepsilon^3\right],$$

$$\gamma_{\rho_2,+} = \frac{m}{n}\left(\varepsilon - \tfrac{1}{4}\varepsilon^2 - \tfrac{5}{16}\varepsilon^3\right), \quad \gamma_{\rho_2,-} = \frac{(m-1)(m+1)}{mn}\left(\varepsilon - \tfrac{1}{4}\varepsilon^2 - \tfrac{5}{16}\varepsilon^3\right),$$

$$\gamma_{\rho_3,+} = \frac{m}{4n}\left(\varepsilon^2 - \tfrac{1}{4}\varepsilon^3\right), \quad \gamma_{\rho_3,-} = -\frac{1}{mn}\left[\varepsilon - \tfrac{1}{4}(m^2 + 1)\varepsilon^2 + \tfrac{1}{16}(m^2 - 5)\varepsilon^3\right],$$

$$\gamma_{\rho_4,+} = \frac{1}{n}\left[\varepsilon + \tfrac{1}{4}(m-2)\varepsilon^2 - \tfrac{1}{16}(m+4)\varepsilon^3\right], \quad \gamma_{\rho_4,-} = \frac{m-1}{mn}\left[\varepsilon + \tfrac{1}{4}(m-1)\varepsilon^2 - \tfrac{1}{16}(m+5)\varepsilon^3\right],$$

$$\gamma_{\rho_5,+} = -\frac{1}{n}\left[\varepsilon - \tfrac{1}{4}(m+2)\varepsilon^2 - \tfrac{1}{16}(m-4)\varepsilon^3\right], \quad \gamma_{\rho_5,-} = -\frac{m+1}{mn}\left[\varepsilon - \tfrac{1}{4}(m+1)\varepsilon^2 + \tfrac{1}{16}(m-5)\varepsilon^3\right]. \tag{37}$$

These correspond to the leading scalar operators in the irreps $RS_{\text{even}}$, $SR_{\text{even}}$, $RR_{\text{even}}$, $TT_{\text{even}}$, $AA_{\text{even}}$ above, respectively. The dimensions of the quadratic in $\phi$ operators at the fixed points $U_{\pm}$ are equal to $\Delta_{\sigma,\rho,\pm} = 2 - \varepsilon + \gamma_{\sigma,\rho,\pm}$. Since $\gamma_{\sigma,+}$ and $\gamma_{\rho_1,\pm}$ are equal to $\varepsilon$ at $n \to \infty$, there should exist a large $n$ expansion (independent of the $\varepsilon$ expansion studied in this section) in which the scaling dimensions of these operators at the corresponding fixed points go to 2 in the infinite-$n$ limit. In the next section, using the analytic bootstrap, we will compute the $\frac{1}{n}$ corrections for $d$ arbitrary. We will see that indeed the $\varepsilon$ and large-$n$ expansions agree in their region of overlapping validity. Our non-perturbative numerical bootstrap results in $d = 3$ below are also consistent with the existence of the large-$n$ limit.

When $m = n$ from (33) we have $R_{nn} = 8(3 - n^2)$ and thus the $\varepsilon$ expansion gives a unitary fixed point for $n^2 \leqslant 3$.[15] For positive integer $n$ this is only satisfied for the uninteresting case $n = 1$.

When $m = n > 1$ and we treat the two $U(n)$ factors as indistinguishable, then the indices $u, v, w$ in (31) take only two values (as opposed to three in the $m \neq n$ case). As a result, the $\rho_{ij}\phi_i\phi_j$ operators decompose into four distinct cases (as opposed to the five in (37)), due to the fact that $RS_{\text{even}}$ and $SR_{\text{even}}$ can no longer be distinguished. The anomalous dimensions of the corresponding operators are real when $n^2 \leqslant 3$.

If we take $m, n$ large with $m/n$ held fixed, then we observe that $\Delta_{\phi,-} = \Delta_{\phi,+}$, $\Delta_{\sigma,-} + \Delta_{\sigma,+} = d$ and $\Delta_{\rho_i,-} = \Delta_{\rho_i,+}$ for $i = 1, \ldots, 5$. Assuming $m, n > 0$ and $n = \alpha m$, then $R_{mn}$ in (32), (33) is positive when $\alpha < 5 - 2\sqrt{6}\sqrt{1 - 1/m^2}$ or $\alpha > 5 + 2\sqrt{6}\sqrt{1 - 1/m^2}$.

---

[15]This holds for $\varepsilon$ infinitesimal. When $\varepsilon$ is finite, this value is expected to change.

If $m$ is assumed large and $n > m$, then we may focus in the region $\alpha \gtrsim \alpha_c = 5 + 2\sqrt{6}$, where the fixed points $U_\pm$ are unitary. The value of $\alpha_c$ has an $\varepsilon$ expansion that follows from (2). As we will see below, the numerical conformal bootstrap provides evidence that for large $m, n$ the fixed points $U_\pm$ either remain unitary down to $\alpha_c = 1$ in $d = 3$, or their non-unitarities are small enough to still allow the bootstrap to produce a kink. Lastly, let us mention that the results in (37) have the same strict double scaling limit, $\alpha = m/n$ fixed with $m$ and $n$ large, with the corresponding results in [19], differing only in subleading $1/n$ corrections. We will see this reflected in one of our plots later (see the discussion around Fig. 8).

## 5 Results in the large-$n$ expansion

In this section we use analytic bootstrap methods as outlined in [19, Sec. 3] to determine scaling dimensions of operators at leading order in $1/n$ as a function of the spacetime dimension $d$. Our basic assumption is that there exist auxiliary Hubbard–Stratonovich fields as leading scalar operators in some irreps. This assumption can be verified a posteriori by means of a comparison with the $\varepsilon$ expansion results of the previous section.

The essential ingredient needed for our application of the analytic bootstrap method is the crossing equation. The four-point function of $\phi$ is written in the form

$$\langle \phi_i(x_1)\phi_j(x_2)\phi_k(x_3)\phi_l(x_4)\rangle = \frac{1}{(x_{12}^2 x_{34}^2)^{\Delta_\phi}} \sum_I P_{ijkl}^I \, \mathcal{G}_I(u,v)\,, \tag{38}$$

where $P_{ijkl}^I$ are the projectors (20), $u, v$ are the usual cross-ratios defined by

$$u = \frac{x_{12}^2 x_{34}^2}{x_{13}^2 x_{24}^2}\,, \qquad v = \frac{x_{14}^2 x_{23}^2}{x_{13}^2 x_{24}^2}\,, \qquad x_{ij} = x_i - x_j\,, \tag{39}$$

and

$$\mathcal{G}_I(u,v) = \sum_{\mathcal{O}_I} c_{\phi\phi\mathcal{O}_I}^2 \, G_{\Delta_{\mathcal{O}_I}, \ell_{\mathcal{O}_I}}(u,v)\,, \tag{40}$$

with $G_{\Delta,\ell}(u,v)$ the usual conformal block [27–29].[16] The crossing equation follows from exchanging operators at $x_2$ and $x_4$ and can we written as

$$\mathcal{G}_I(u,v) = M_{IJ} \left(\frac{u}{v}\right)^{\Delta_\phi} \mathcal{G}_J(v,u)\,, \tag{41}$$

where the explicit form of the $12 \times 12$ matrix $M_{IJ}$ is easy to work out and is included in an ancillary file.

Imposing that the leading spin-two operator in the irrep $S_{\text{even}}$ is the stress-energy tensor with dimension $d$, and that the leading spin-one operators in the irreps $RS_{\text{odd}}$ and $SR_{\text{odd}}$ are conserved currents with dimensions $d - 1$, we may determine the dimensions of operators at leading order in $1/n$ in each of the $U_\pm$ fixed points. Here we report only the leading scalar operators. We have included operators of higher spin in an ancillary file. First let us define $\mu = d/2$ and

$$\eta_1 = \frac{(\mu-2)\Gamma(2\mu-1)\sin(\pi\mu)}{\pi\Gamma(\mu)\Gamma(\mu+1)}\,. \tag{42}$$

---

[16] Our conventions for the normalization of the conformal block are those of [30].

At $U_+$ we find, at leading order in $1/n$,

$$
\begin{aligned}
\Delta_{\phi,+} &= \mu - 1 + \frac{m}{2}\frac{\eta_1}{n} & &\stackrel{\text{3d}}{=} \frac{1}{2} + \frac{2m}{3\pi^2 n}, \\
\Delta_{S,+} &= 2 - \frac{2(\mu-1)(2\mu-1)m}{2-\mu}\frac{\eta_1}{n} & &\stackrel{\text{3d}}{=} 2 - \frac{16m}{3\pi^2 n}, \\
\Delta_{RS,+} &= 2 - \frac{2(\mu-1)^2 m}{2-\mu}\frac{\eta_1}{n} & &\stackrel{\text{3d}}{=} 2 - \frac{4m}{3\pi^2 n}, \\
\Delta_{SR,+} &= 2\Delta_{\phi+} + \frac{\mu m}{2-\mu}\frac{\eta_1}{n} & &\stackrel{\text{3d}}{=} 1 + \frac{16m}{3\pi^2 n}, \\
\Delta_{RR,+} &= 2\Delta_{\phi+} & &\stackrel{\text{3d}}{=} 1 + \frac{4m}{3\pi^2 n}, \\
\Delta_{TT,+} &= 2\Delta_{\phi+} + \frac{\mu}{2-\mu}\frac{\eta_1}{n} & &\stackrel{\text{3d}}{=} 1 + \frac{4(m+3)}{3\pi^2 n}, \\
\Delta_{AA,+} &= 2\Delta_{\phi+} - \frac{\mu}{2-\mu}\frac{\eta_1}{n} & &\stackrel{\text{3d}}{=} 1 + \frac{4(m-3)}{3\pi^2 n}.
\end{aligned}
\tag{43}
$$

At $U_-$ and again at leading order in $1/n$ we find

$$
\begin{aligned}
\Delta_{\phi,-} &= \mu - 1 + \frac{(m-1)(m+1)}{2m}\frac{\eta_1}{n} & &\stackrel{\text{3d}}{=} \frac{1}{2} + \frac{2(m-1)(m+1)}{3\pi^2 mn}, \\
\Delta_{S,-} &= 2\Delta_{\phi-} + \frac{\mu(4\mu-5)(m-1)(m+1)}{(2-\mu)m}\frac{\eta_1}{n} & &\stackrel{\text{3d}}{=} 1 + \frac{16(m-1)(m+1)}{3\pi^2 mn}, \\
\Delta_{RS,-} &= 2 - \frac{2}{2-\mu}\Big[(\mu-1)^2 m - \frac{4\mu^2-6\mu+1}{m}\Big]\frac{\eta_1}{n} & &\stackrel{\text{3d}}{=} 2 - \frac{4(m-2)(m+2)}{3\pi^2 mn}, \\
\Delta_{SR,-} &= 2\Delta_{\phi-} + \frac{\mu(m-1)(m+1)}{(2-\mu)m}\frac{\eta_1}{n} & &\stackrel{\text{3d}}{=} 1 + \frac{16(m-1)(m+1)}{3\pi^2 mn}, \\
\Delta_{RR,-} &= 2\Delta_{\phi-} - \frac{\mu}{(2-\mu)m}\frac{\eta_1}{n} & &\stackrel{\text{3d}}{=} 1 + \frac{4(m-2)(m+2)}{3\pi^2 mn}, \\
\Delta_{TT,-} &= 2\Delta_{\phi-} + \frac{\mu(m-1)}{(2-\mu)m}\frac{\eta_1}{n} & &\stackrel{\text{3d}}{=} 1 + \frac{4(m-1)(m+4)}{3\pi^2 mn}, \\
\Delta_{AA,-} &= 2\Delta_{\phi-} - \frac{\mu(m+1)}{(2-\mu)m}\frac{\eta_1}{n} & &\stackrel{\text{3d}}{=} 1 + \frac{4(m-4)(m+1)}{3\pi^2 mn}.
\end{aligned}
\tag{44}
$$

We have checked that the $\mu$-dependent results are consistent with (35), (36) and (37) when expanded in $\varepsilon$ with $\mu = 2 - \varepsilon/2$. To our knowledge the large-$n$ results presented here are new.

# 6 Numerical bootstrap results

We start this section by noting that in the various plots we will label bounds for $U(m) \times U(n)$ theories by $U_{m,n}$ for brevity. We emphasize here that, since our bootstrap bounds are obtained with the four-point function of $\phi$ only, they apply to theories with $U(m) \times U(n)$, $[U(m) \times U(n)]/U(1)$, and $SU(m) \times SU(n)$ global symmetry.[17] For the case $m = n$ we may treat the two $U(n)$ factors as distinguishable or indistinguishable and we will explicitly mention our choice in context. With the latter choice those are bounds for theories with $U(n)^2 \rtimes \mathbb{Z}_2$ global symmetry, and we will label these by $\widehat{U}_{n,n}$ in the corresponding plots.[18] Whenever squares and circles appear in

---

[17] $[U(m) \times U(n)]/U(1)$ bounds are necessarily weaker than corresponding $U(m) \times U(n)$ bounds, but the two may also coincide.

[18] $U(n) \times U(n)$ bounds are necessarily weaker than corresponding $U(n)^2 \rtimes \mathbb{Z}_2$ bounds, but the two may also coincide.

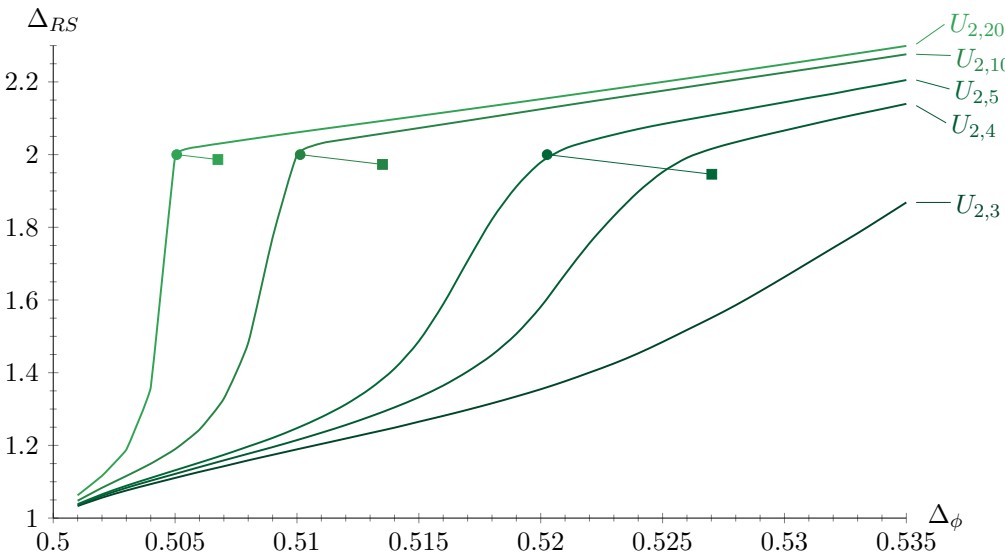

Figure 2: Upper bound on the dimension of the first scalar $RS$ operator in the $\phi \times \phi$ OPE as a function of the dimension of $\phi$. Areas above the curves are excluded in the corresponding theories. The locations of the fixed points as predicted by the large-$n$ results (43) and (44) for $n = 5, 10, 20$ are also given as squares and circles for the $U_+$ and $U_-$ fixed points, respectively. (The lines between squares and circles are added to illustrate that the corresponding fixed points have the same symmetry.)

the plots, these correspond to the location of the $U_+$ and $U_-$ fixed points of the corresponding Lagrangian theories, respectively, according to the large-$n$ results (43) and (44). Straight lines connecting squares and circles are added to illustrate that the connected fixed points correspond to the same $m, n$. It is emphasized that the fixed points $U_\pm$ are added in plots without examining the issue of their existence as unitary fixed points. The parameters used in the numerics are discussed in Appendix C. The crossing equations are included in an ancillary file.

In Fig. 2 we present bounds for the dimension of the first scalar operator in the $RS$ irrep as a function of the dimension of $\phi$. We work with $U(2) \times U(n)$ theories, but similar behavior is seen in bounds at higher $m$. These bounds display sharp kinks at large $n$. Using our analytic large-$n$ results of the previous section, we see that the $U_-$ fixed points are responsible for these kinks.[19]

An interesting question is whether there exists a kink in the $U(2) \times U(2)$ theory. The corresponding $\Delta_{RS}$ bound is seen in Fig. 3. There we see that the $\Delta_{RS}$ bound for the $U(2) \times U(2)$ theory is much stronger than the $\Delta_{RS}$ bound for the $U(2) \times U(3)$ theory. The difference is much more significant than that between the $U(3) \times U(3)$ and $U(3) \times U(4)$ theories, which are also shown in Fig. 3 for comparison. This indicates that the $U(2) \times U(3)$ theory is sensitive to a potential fixed point which has no extension to the $U(2) \times U(2)$ theory. Indeed, a kink is clearly forming in the $U(2) \times U(3)$ bound, while no kink at all is present in the $U(2) \times U(2)$ bound.[20] Comparing with Fig. 2, it seems plausible that the kink in the $U(2) \times U(3)$ bound in Fig. 3 is due to the corresponding $U_-$ fixed point. Note that [6] estimated $n^+(2) = 4.373(18)$ in $d = 3$,

---

[19]That being said, we observed that extracting the spectrum at e.g. the $U(2) \times U(20)$ kink, there was no sign of the "$\phi^4$"-type singlet with dimension $\Delta_S = 2 + O(1/n)$ expected from the large-$n$ description. A similar observation was made for the bound corresponding to the $O(2) \times O(10)$ anti-chiral fixed point in [19]. Let us also mention that operators have been known to be missing from the extracted spectrum even in theories which are under very good control in the numerical bootstrap, such as the Ising model. For example, in Figure 11 of [31], while the second and fourth $\mathbb{Z}_2$-odd spin-0 operators are captured by the numerics and agree with perturbative estimates, the third operator is not seen.

[20]We have obtained the $U(2) \times U(2)$ $\Delta_{RS}$ bound up to $\Delta_\phi = 0.7$ and no kink is seen.

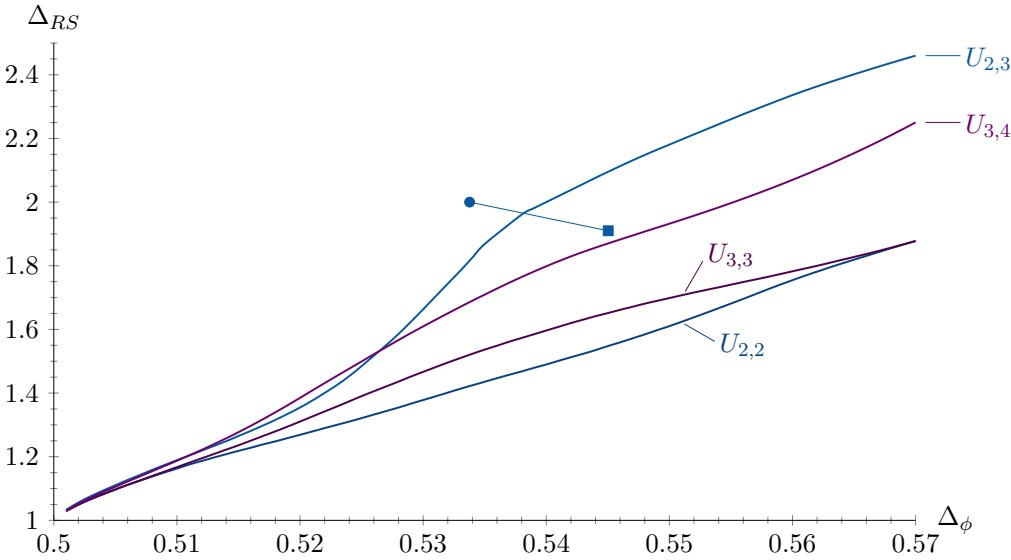

Figure 3: Upper bound on the dimension of the first scalar *RS* operator in the $\phi \times \phi$ OPE as a function of the dimension of $\phi$. Areas above the curves are excluded in the corresponding theories. The $U(2) \times U(2)$ and $U(3) \times U(3)$ bounds are obtained assuming the two factors of the global symmetry group are distinguishable. The location of the fixed points $U_+$ (square) and $U_-$ (circle) as predicted by the large-$n$ results (43) and (44) for the $U(2) \times U(3)$ theory are also shown. (The line in between is added to illustrate that the corresponding fixed points have the same symmetry.)

which if correct would imply that both $U(2) \times U(4)$ and $U(2) \times U(3)$ kinks in Fig. 2 correspond to non-unitary fixed points.

Our conclusion from the $\Delta_{RS}$ bounds is that the $U(2) \times U(2)$ theory does not have a unitary $U_-$ fixed point. This is consistent with expectations from perturbative methods [5, 6]. We do stress, however, that this does not in principle exclude some other fixed point of a different type for this symmetry (e.g. a fixed point inaccessible through standard perturbation theory); see [9], [7] and also our discussion below pertaining to Fig. 6.

In Fig. 4 we plot bounds for the dimension of the first scalar operator in the *SR* irrep as a function of the dimension of $\phi$, again for $U(2) \times U(n)$ theories for various values of $n$. Here we do not see kinks as sharp as those of Fig. 2, but at large $n$ we do observe changes in slope that are saturated by the $U_+$ fixed point. This is more clear for the $U(2) \times U(20)$ theory in Fig. 5, where we plot bounds on the dimensions of the leading operators in all five scalar non-singlet irreps that appear in the $\phi \times \phi$ OPE. The blue circle in each plot in Fig. 5 corresponds to the $O(80)$ model, which saturates the $\Delta_{RR}$, $\Delta_{TT}$ and $\Delta_{AA}$ bounds. Note that the $\Delta_{TT}$ bound in Fig. 5 is saturated both by the $O(80)$ model and $U_-$ for different values of $\Delta_\phi$, without sharp kinks in either case.

In Fig. 6 we plot bounds on the leading scalar non-singlet operators in the case of $U(2) \times U(2)$ symmetry with the $O(8)$ fixed point marked in blue. As we have already mentioned, the $\Delta_{RS}$ bound does not display a kink, which is interpreted as the absence of a unitary $U_+$ fixed point for $m = n = 2$. However, the $\Delta_{AA}$ bound displays a kink around $\Delta_\phi = 0.53$. This kink was first observed in the $O(2) \times O(4)$ studies of [7, 19] (see [7, Fig. 3] and [19, Fig. 3]).[21]

In Fig. 7 we plot bounds on the leading scalar non-singlet operators for 3D CFTs with

---

[21]The bounds in Fig. 6 are valid for 3D CFTs with either $U(2) \times U(2)$ or $[U(2) \times U(2)]/U(1) \simeq SU(2) \times SU(2) \times U(1) \simeq SO(4) \times SO(2)$ global symmetry. The fact that the $\Delta_{AA}$ bound in Fig. 6 coincides with a bound obtained for 3D CFTs with $O(4) \times O(2)$ global symmetry means that 3D CFTs with $U(2) \times U(2)$ symmetry, should any exist, lie in the allowed region of the $\Delta_{AA}$ bound.

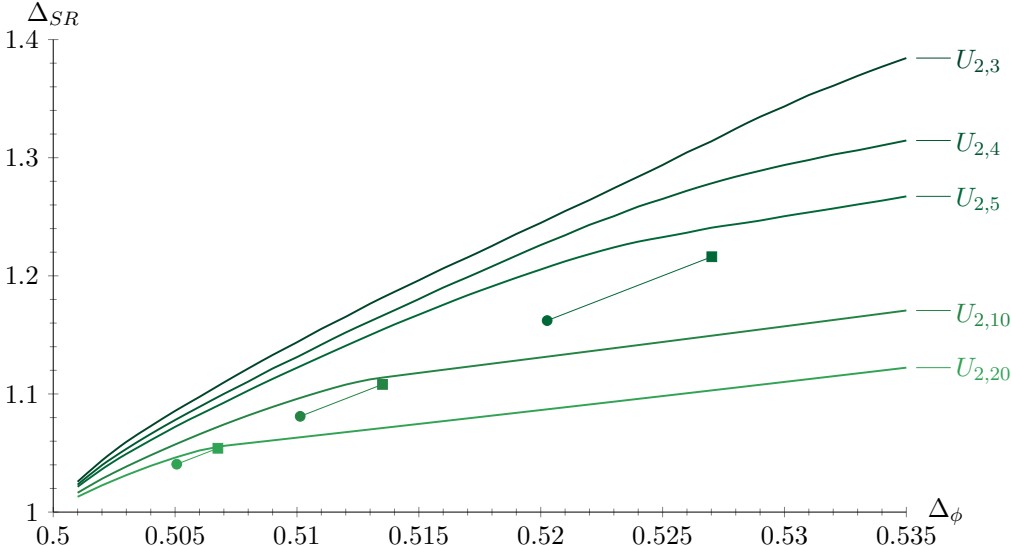

Figure 4: Upper bound on the dimension of the first scalar *SR* operator in the $\phi \times \phi$ OPE as a function of the dimension of $\phi$. Areas above the curves are excluded in the corresponding theories. The locations of the fixed points as predicted by the large-$n$ results (43) and (44) for $n = 5, 10, 20$ are also given as squares and circles for the $U_+$ and $U_-$ fixed points, respectively. (The lines between squares and circles are added to illustrate that the corresponding fixed points have the same symmetry.)

$U(3) \times U(3)$ symmetry. The $O(18)$ fixed point is marked in blue. Here we observe no kinks at all, indicating that the $U_\pm$ fixed points of the $\varepsilon$ expansion do not appear to survive as unitary fixed points for $m = n = 3$.

From our results so far it appears that the $U_\pm$ fixed points of the $\varepsilon$ expansion are not of importance for the chiral phase transition of two- and three-flavor massless QCD. Using our setup we can extend this question to QCD with (parametrically) many massless flavors. In Fig. 8 we present bounds on the dimension of the leading scalar operator in the *RSSR* irrep, where we assume $m = n$ and that the two $U(n)$ factors are indistinguishable. We have also obtained $\Delta_{RS}$ and $\Delta_{SR}$ bounds assuming that the two $U(n)$ factors are distinguishable, which are identical between themselves and differ slightly from the ones shown in Fig. 8 only for $n = 2, 3$. For large $n$ the $\Delta_{RS}, \Delta_{SR}$ bounds coincide with the corresponding $\Delta_{RSSR}$ bound of Fig. 8. As we observe, despite the absence of a kink for low $n$ a kink is clearly seen at large $n$.[22] From Fig. 9 we expect this kink to be due to the $U_-$ fixed point.

We note that the $m = n = 100$ bound in Fig. 8 is essentially identical to the one in [33, Fig. 12]. This explains the origin of the kink seen in that bound. Note that the operator $Z_{ij}^{ab}$ in that work is equivalent (when $a, b = 1, 2$) to a bifundamental operator $\phi_{ij}$ of $O(n) \times O(n)$ in the case where the $O(n)$ factors are indistinguishable. Then, following our discussion at the end of section 4, it becomes clear why these bounds can coincide at large $n$.

The $\widehat{U}_{1000,1000}$ kink in Fig. 9 (which is identical to the $\Delta_{RS}$ kink in Fig. 10) is also relevant for the possible existence of a $U(m) \times U(n)$ 3D CFT at $m, n$ large with $m/n$ fixed and close to 1. As we discussed at the bottom of section 4, when $m, n$ are both large but $m/n$ is sufficiently small, the $U_\pm$ fixed points are unitary. Since the kink in Fig. 9 survives as we increase the ratio $m/n$ towards 1, we may conclude that either the $U_\pm$ fixed points survive as unitary fixed points in that case, or the possible non-unitarities are too small to stop the kink from forming.

---

[22]A similar situation has been encountered in the bootstrap of four-point functions of scalar adjoint operators in 3D CFTs with $SU(N)$ global symmetry [32, Fig. 2].

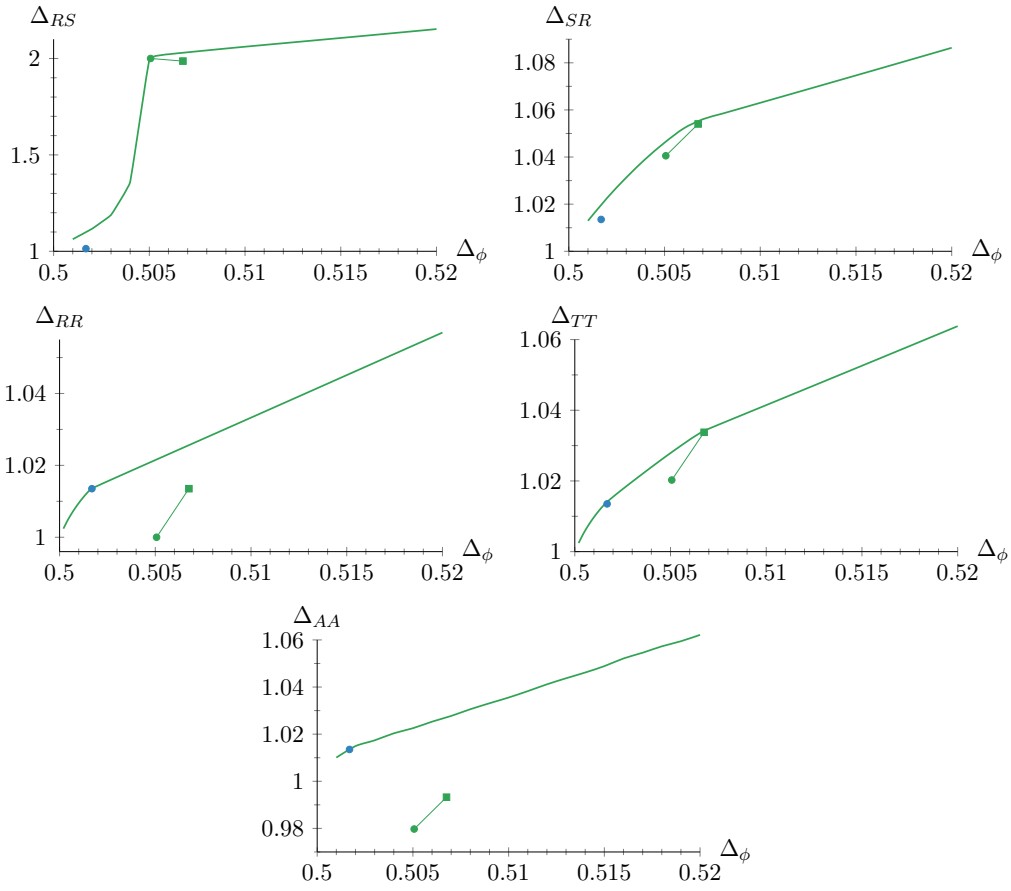

Figure 5: Upper bounds on the dimensions of the first scalar operators in the various irreps that appear in the $\phi \times \phi$ OPE as functions of the dimension of $\phi$ for $U(2) \times U(20)$ global symmetry. The blue circle marks the location of the $O(80)$ fixed point, which for all plots in the vertical axis is given by the dimension of the leading scalar two-index traceless symmetric operator in that theory. The $U_\pm$ fixed points are also shown as green circles ($U_-$) and squares ($U_+$), connected by a thin line to indicate that they have the same global symmetry. Areas above the curves are excluded.

Therefore, QCD[23] with 1000 massless flavors may undergo a (near) second order chiral phase transition due to the presence of the $U_+$ fixed point. It would be interesting to compute the value of $n^+(1000)$ in the $\varepsilon$ expansion using the results of [6] and see which of the two pictures it corroborates. If the fixed point is indeed non-unitary this could give us a quantitative measure[24] of the sensitivity to non-unitarities in the bootstrap.

Let us comment on Fig. 10. We have already discussed the kink in the $\Delta_{RS}$ bound. We additionally observe that the $\Delta_{RR}, \Delta_{TT}, \Delta_{AA}$ bounds essentially coincide. This coincidence is also seen in the large-$n$ results (43) and (44) when we take $m$ large and equal to $n$, although taking $m$ large in those results is not justified. We have also compared with the large-$m, n$ $O(m) \times O(n)$ results of [19] and we find the same set of scaling dimensions.

In Fig. 11 we show the $RS$ exclusion bound for $m = 3$ fixed and increasing $n$. At large $n$ these bounds have kinks that are saturated by the corresponding $U_+$ fixed points. For $n = 3, 4$ no kinks are found up to $\Delta_\phi = 0.54$, and so we expect $n^+(3) > 4$. Nevertheless, the $U(3) \times U(3)$

---

[23]Or, more appropriately, Yang–Mills theory with a sufficiently large number of colors if one wants it to be confining.

[24]Since one may tune the size of the non-unitarity by tuning $1/n$.

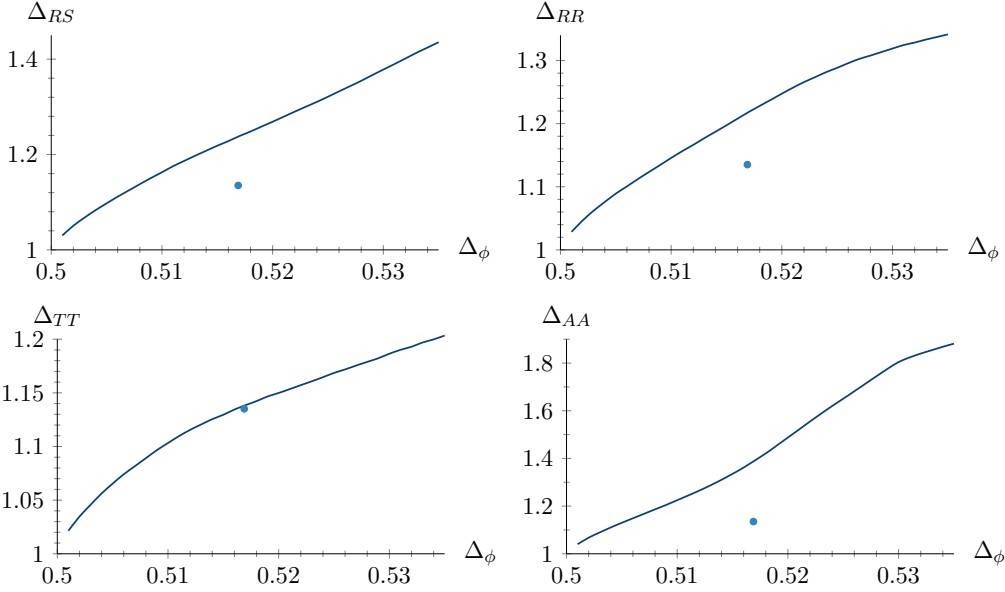

Figure 6: Upper bounds on the dimensions of the first scalar operators in the various irreps that appear in the $\phi \times \phi$ OPE as functions of the dimension of $\phi$ for $U(2) \times U(2)$ global symmetry, where the two $U(2)$ factors are considered distinguishable. The blue circle marks the location of the $O(8)$ fixed point, which for all plots in the vertical axis is given by the dimension of the leading scalar two-index traceless symmetric operator in that theory. Areas above the curves are excluded. The $\Delta_{SR}$ bound is identical to the $\Delta_{RS}$ one.

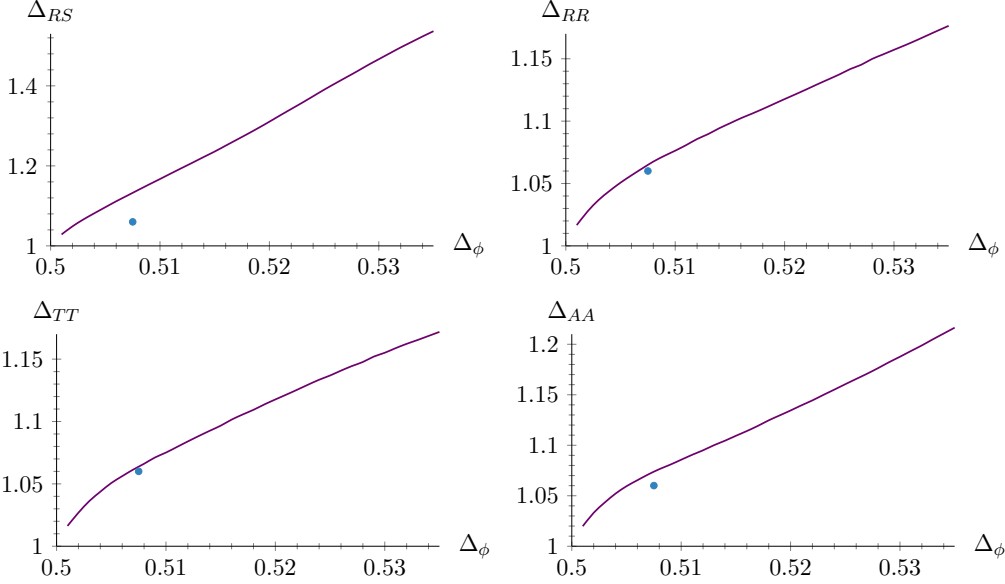

Figure 7: Upper bounds on the dimensions of the first scalar operators in the various irreps that appear in the $\phi \times \phi$ OPE as functions of the dimension of $\phi$ for $U(3) \times U(3)$ global symmetry, where the two $U(3)$ factors are considered distinguishable. The blue circle marks the location of the $O(18)$ fixed point, which for all plots in the vertical axis is given by the dimension of the leading scalar two-index traceless symmetric operator in that theory. Areas above the curves are excluded. The $\Delta_{SR}$ bound is identical to the $\Delta_{RS}$ one.

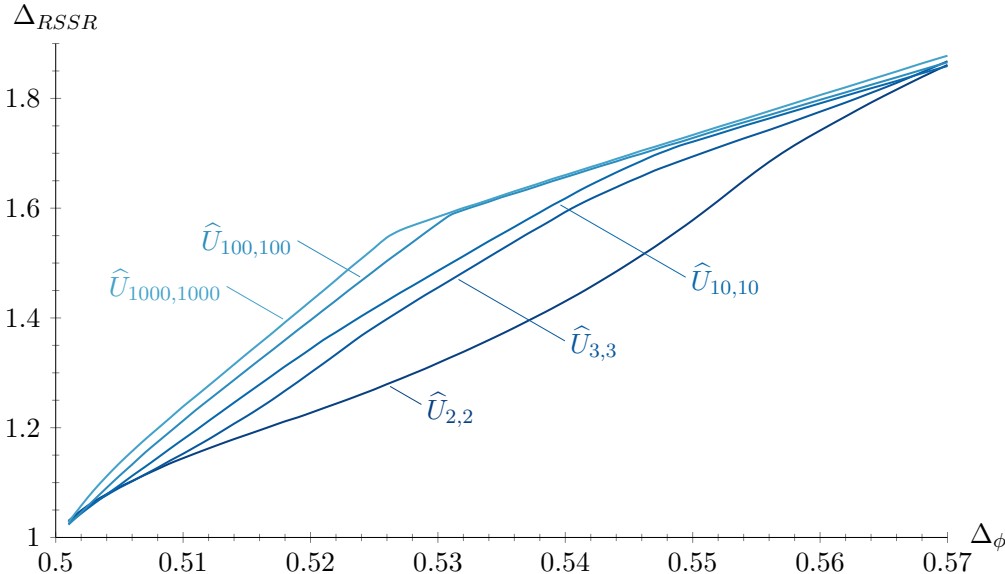

Figure 8: Upper bound on the dimension of the first scalar *RSSR* operator in the $\phi \times \phi$ OPE as a function of the dimension of $\phi$. The global symmetry group for the various bounds here is $U(n)^2 \rtimes \mathbb{Z}_2$. Areas above the curves are excluded in the corresponding theories.

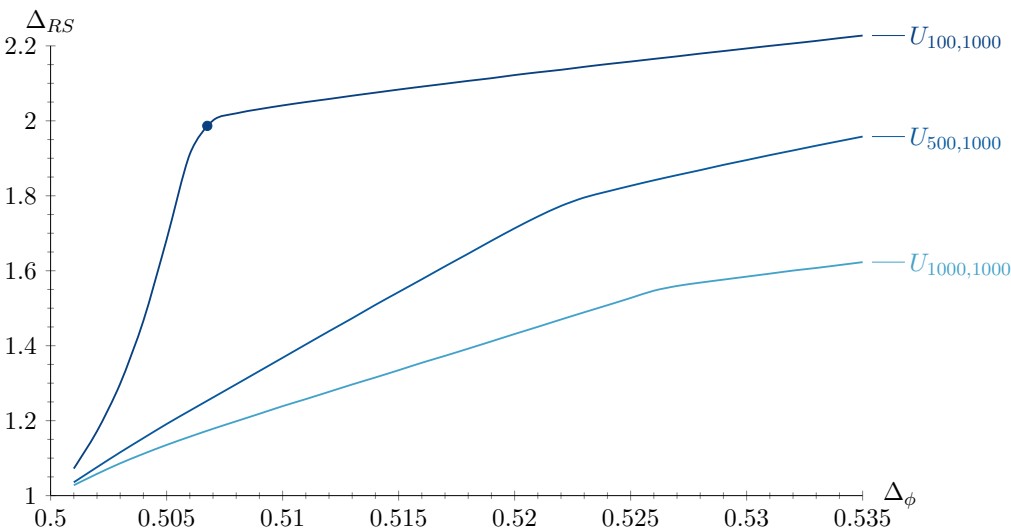

Figure 9: Upper bound on the dimension of the first scalar *RS* operator in the $\phi \times \phi$ OPE as a function of the dimension of $\phi$. The $U_{1000,1000}$ bound has been obtained treating the two $U(1000)$ factors as distinguishable; however, it is identical to the $\widehat{U}_{1000,1000}$ bound in Fig. 8. Areas above the curves are excluded in the corresponding theories. The location of the $U(100) \times U(1000)$ $U_-$ fixed point as predicted by the large-$n$ results (44) theory is also shown. The $U(100) \times U(1000)$ $U_+$ fixed point is at the same location.

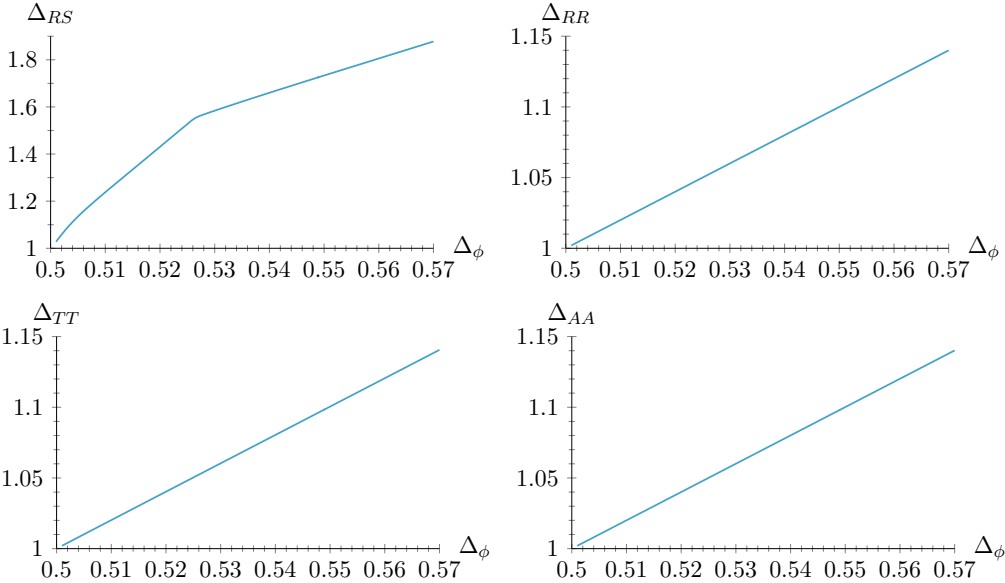

Figure 10: Upper bounds on the dimensions of the first scalar operators in the various irreps that appear in the $\phi \times \phi$ OPE as functions of the dimension of $\phi$ for $U(1000) \times U(1000)$ global symmetry, where the two $U(1000)$ factors are considered distinguishable. Areas above the curves are excluded. The $\Delta_{SR}$ bound is identical to the $\Delta_{RS}$ one.

bound does have a kink as we see in Fig. 12. This kink appears to be unrelated to the $U_+$ fixed point of the $\varepsilon$ expansion. As we see from Fig. 13 this kink disappears as we increase $n$ with $m = 3$ fixed. Its relevance for the nature of the chiral phase transition of QCD remains to be seen, but its existence opens the possibility that it may be second order. We stress, however, that this fixed point need not necessarily be due to a multi-scalar theory, but may also be due to a gauge theory or a Gross–Neveu–Yukawa (GNY) theory.[25]

Before concluding, let us discuss a few more plots that present features which may be of interest to the bootstrap in general. In Fig. 14 we present bounds on operators in various representations of the global symmetry for the $U(3) \times U(20)$ CFTs. There are a couple of features that stand out. Firstly, the exclusion bound for the $l = 1$ $RR$ operator is almost exactly saturated by the $U_-$ fixed point, even though the plot is essentially a straight line, absent of even the mildest feature. Secondly, the bound on the $l = 0$ $TT$ operator is also saturated very well by the $U_+$ fixed point, albeit in this case it does have a very minor feature (a very minor change of slope). In fact, throughout this work, we found that the $TT$ bound was always saturated very well by the analytic predictions despite only having a very weak feature. These examples show that a lot of mundane looking bootstrap plots may actually be much richer than initially thought. To reiterate, we saw explicitly that the $TT$ bound is saturated by not just one, but two fixed points. Lastly, in Fig. 15 we plot the $TA$ exclusion bound as a function of increasing spin. For $l = 1$ we see that the bound is saturated by the $U_-$ fixed point. Then, at $l = 3$ the bound is almost saturated by three distinct fixed points, again despite having no feature. As the spin is further increased the agreement becomes progressively worse, which may be due to loss of constraining power.

---

[25]As it is located at values of $\Delta_\phi$ larger than the typical ones expected for multi-scalar theories (0.5+corrections). We remind the reader that, for example in GNY theories, one obtains a correction to the anomalous dimension of $\phi$ an order earlier in perturbation theory (at one loop instead of two loops as in a pure scalar field theory).

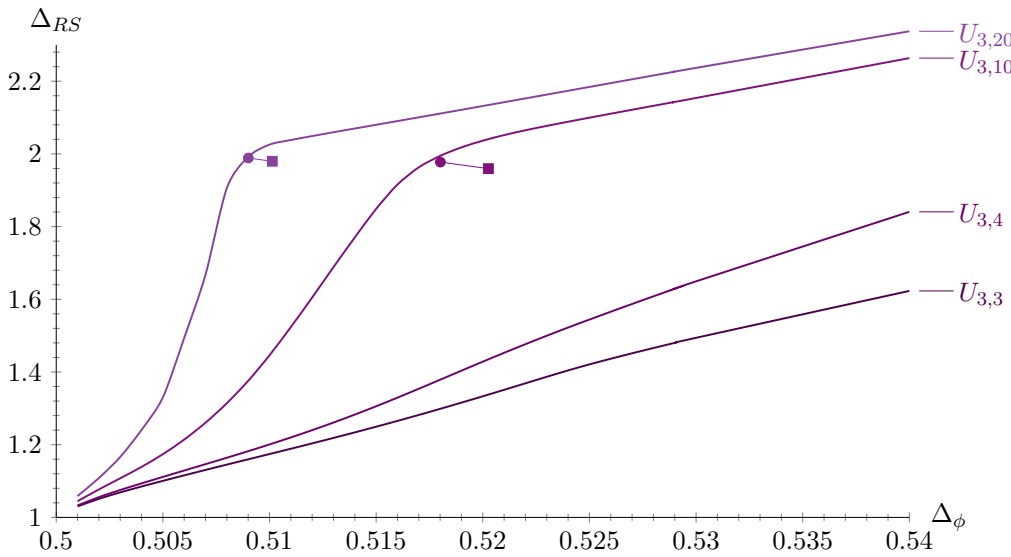

Figure 11: Upper bound on the dimension of the first scalar *RS* operator in the $\phi \times \phi$ OPE as a function of the dimension of $\phi$. Areas above the curves are excluded in the corresponding theories. The locations of the fixed points as predicted by the large-$n$ results (43) and (44) for $n = 3, 4, 10, 20$ are also given as squares and circles for the $U_+$ and $U_-$ fixed points, respectively. The lines between squares and circles (for $n = 10, 20$) are added to illustrate that the corresponding fixed points have the same symmetry. These plots were run with parameters "*D*" in Appendix C.

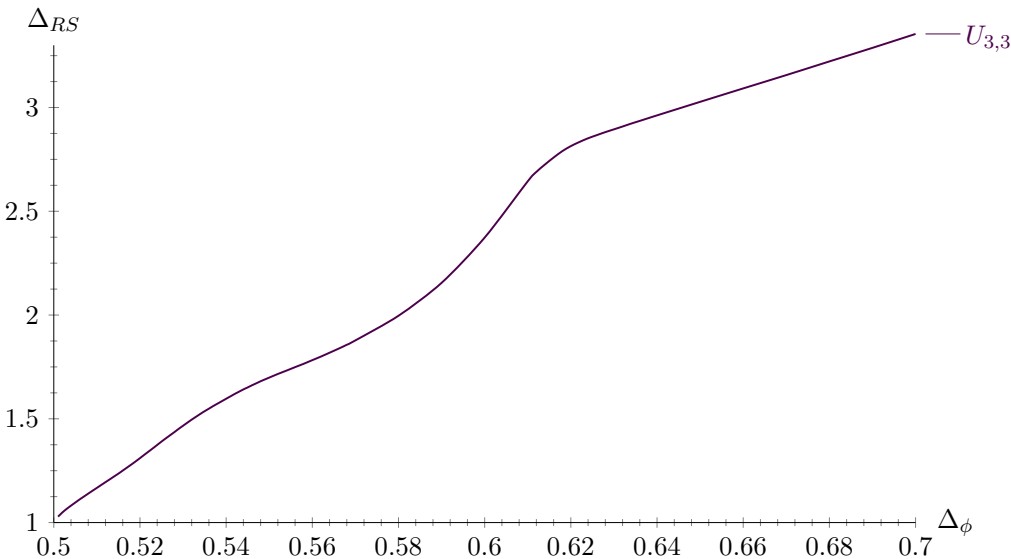

Figure 12: Upper bound on the dimension of the first scalar *RS* operator in the $\phi \times \phi$ OPE as a function of the dimension of $\phi$. The area above the curves is excluded.

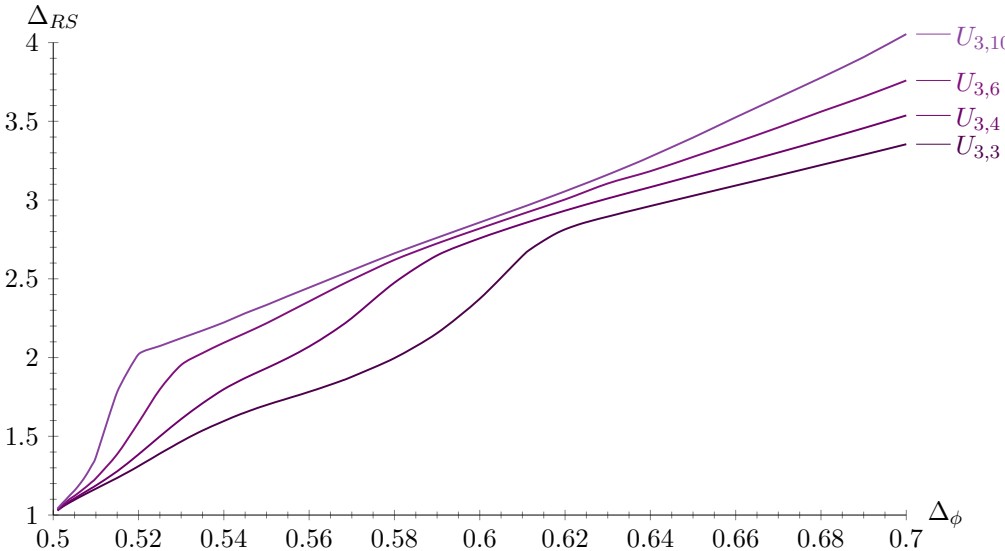

Figure 13: Upper bound on the dimension of the first scalar $RS$ operator in the $\phi \times \phi$ OPE as a function of the dimension of $\phi$. Areas above the curves are excluded in the corresponding theories.

# 7 Discussion and future directions

In the present work we performed a comprehensive study of $U(m) \times U(n)$ symmetric CFTs. This included perturbative computations in the $\varepsilon = 4 - d$ and large-$n$ expansions, and non-perturbative computations with the numerical conformal bootstrap in $d = 3$. When $m = n$ we analyzed the cases where the two $U(n)$ factors in $U(n) \times U(n)$ are considered distinguishable or indistinguishable. Our study was motivated both by phenomenological applications to the chiral phase transition of massless QCD, as well as purely field theoretical considerations.

For the phenomenologically interesting values of $m = n = 2$ for two-flavor massless QCD, we found that as $n$ is lowered from $n = 3$ to $n = 2$ with $m = 2$ fixed (see Fig. 3), there is a large change in the corresponding bootstrap bound. More specifically, the bound becomes much stronger (i.e. it excludes much more of parameter space) and lacks a kink that existed for larger values of $n$ (see Fig. 2). This could be explained by the disappearance of a unitary fixed point as we lower the value of $n$, such that the bootstrap may then exclude the region in parameter space originally occupied by that fixed point. If so, whatever fixed point was responsible for the feature in the $U(2) \times U(n)$ exclusion bounds for large $n$, cannot exist for $U(2) \times U(2)$. We stress, though, that this does not exclude the possibility of some other $U(2) \times U(2)$ fixed point. For example, novel fixed points with $O(n) \times O(2)$ symmetry were reported in [34] (remember that $O(4) \times O(2) \sim SU(2) \times SU(2) \times U(1)$). Indeed a kink is observed in the $\Delta_{AA}$ bound of Fig. 6, which may be attributed to a unitary CFT unrelated to the fixed points found in the $\varepsilon$ expansion. The $\Delta_{AA}$ bound of Fig. 6 is identical to a bound obtained for 3D CFTs with $O(4) \times O(2)$ global symmetry [7, 8, 19]. We note that recently [10] found the $O(4) \times O(2)$ transition to be first order.

For $m = n = 3$, a case relevant for the chiral phase transition of three-flavor massless QCD, we also observe a pronounced kink in our bootstrap bound; see Fig. 12. Therefore, our work produces evidence that this transition may be second order.

On the field theoretical side, we observed that computations in the large-$n$ limit provided very accurate predictions for the scaling dimensions of numerous operators; see e.g. Fig. 5. Additionally, we found that large-$n$ results saturated bootstrap bounds, even in the complete ab-

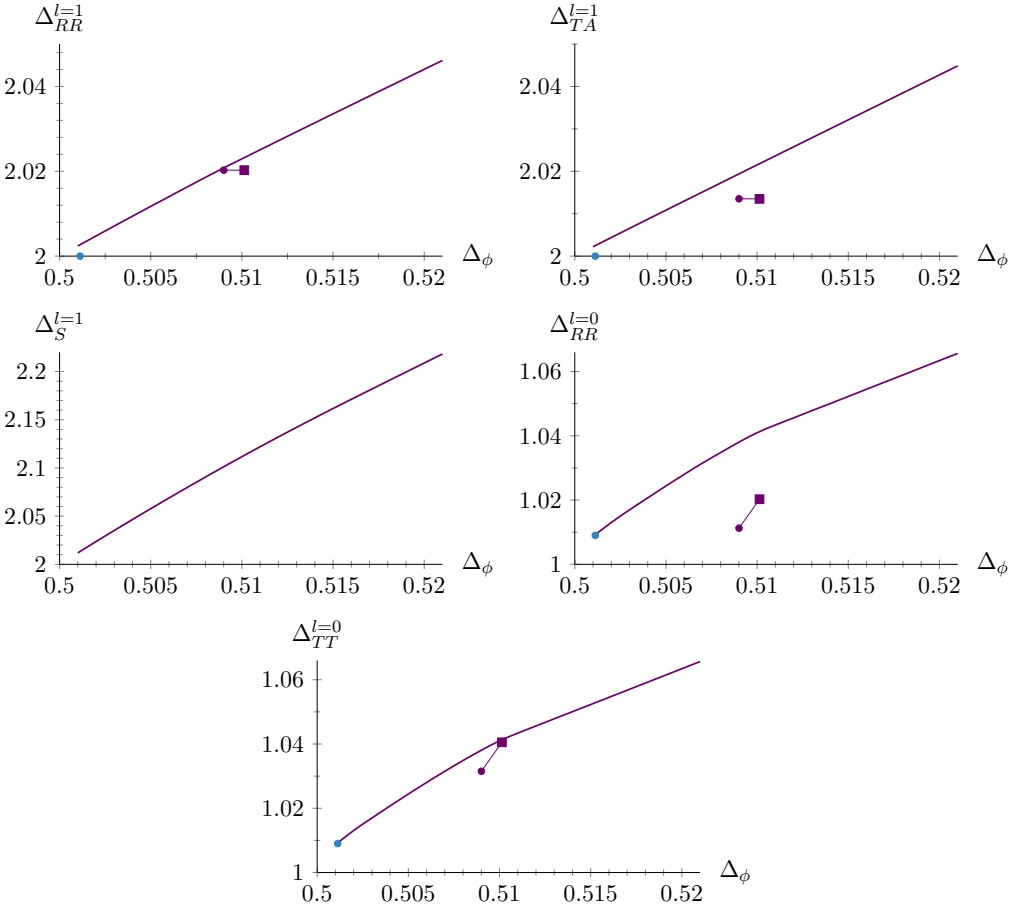

Figure 14: Upper bound on the dimension of various operators in the $\phi \times \phi$ OPE as a function of the dimension of $\phi$ for $m = 3$ and $n = 20$. The spin of each operator is labeled by a superscript. Areas above the curves are excluded in the corresponding theories. The locations of the fixed points as predicted by the large-$n$ results (43) and (44) are also given as squares and circles for the $U_+$ and $U_-$ fixed points, respectively. The lines between squares and circles are added to illustrate that the corresponding fixed points have the same symmetry. The blue circle marks the location of the $O(120)$ fixed point. The first two plots were run with parameters "$\mathcal{B}$" and the rest with parameters "$\mathcal{C}$" in Appendix C.

sence of kinks. Another interesting observation is that in Fig. 9 the unitary $m = 100, n = 1000$ fixed point seems to evolve into the $m = n = 1000$ fixed point as $m$ is increased. One interpretation of this is that for large $m, n$ a unitary $U(m) \times U(n)$ CFT exists even when $m/n \to 1$. An alternative interpretation is that in the limit $m/n \to 1$ with $n \to \infty$ the non-unitarities become suppressed enough for the bootstrap bound to display a kink. Note that the double scaling limit reported in this work also exists in the results of [19].

Given our results, it would be interesting to extend the existing perturbative data available for these theories. More precise perturbative predictions could allow us to follow theories from infinitesimal values of the control parameter to the physically interesting values (e.g. $\varepsilon = 1$ or $m = n = 2, 3$). For example, in [31] in the case of the Ising model within the context of the $\varepsilon$ expansion, the perturbative data was found, a posteriori, to be a very accurate description of the full non-perturbative theory (at least in the absence of operator mixing). The extension of results for $U(m) \times U(n)$ theories in the $\varepsilon$ expansion to higher order in perturbation theory and

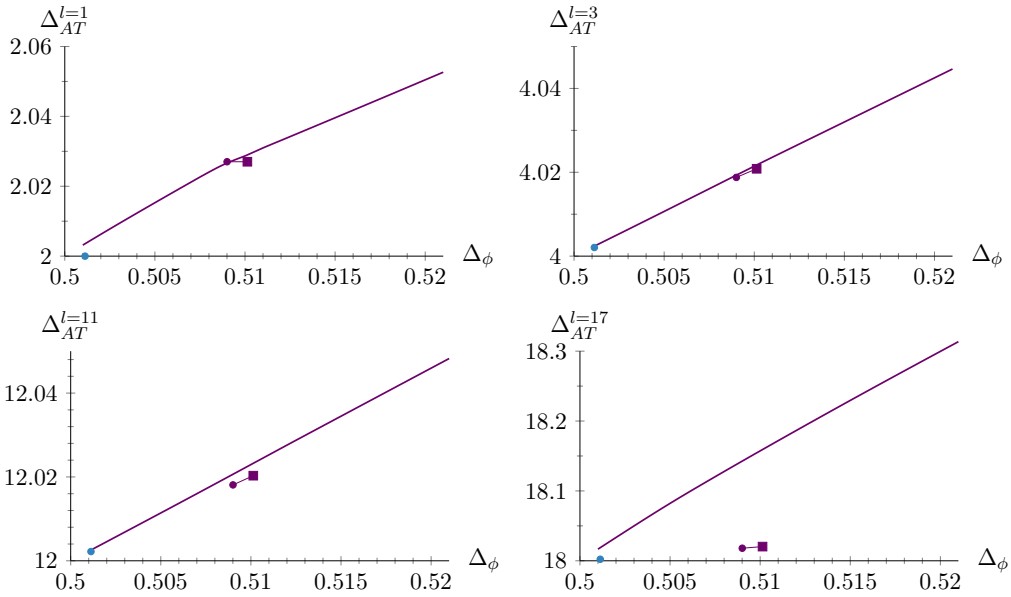

Figure 15: Upper bound on the dimension of $AT$ operators of various spins in the $\phi \times \phi$ OPE as a function of the dimension of $\phi$ for $m = 3$ and $n = 20$. The spin of each operator is labeled by a superscript. Areas above the curves are excluded in the corresponding theories. The locations of the fixed points as predicted by the large-$n$ results (43) and (44) are also given as squares and circles for the $U_+$ and $U_-$ fixed points, respectively. The lines between squares and circles are added to illustrate that the corresponding fixed points have the same symmetry. The blue circle marks the location of the $O(120)$ fixed point. These plots were run with parameters "$\mathcal{B}$" in Appendix C.

more operators is possible with the results of [24]. Extension of our large $n$ results to higher orders would also be desirable, especially seeing the usefulness of even leading order results, when used in conjunction with the numerical bootstrap. On the numerical side, we would like to be able to precisely pinpoint the values of $m$ and $n$ which separate the different regimes of fixed points (which we discussed in section 2).

## Acknowledgments

We thank R. Pisarski for insightful correspondence, reading through the manuscript and pointing out relevant literature. We also thank J. Henriksson for helpful conversations on the analytic bootstrap. Additionally, we are grateful to three anonymous referees whose reports helped improve this manuscript. Some computations in this paper have been performed with the help of *Mathematica* with the packages xAct [36] and xTras [37].

**Funding information** A. S. is funded by the Royal Society under grant URF\R1\211417 "Advancing the Conformal Bootstrap Program in Three and Four Dimensions." The research work of SRK received funding from the European Research Council (ERC) under the European Union's Horizon 2020 research and innovation programme (grant agreement no. 758903). The numerical computations in this work have used King's College London's Rosalind and CREATE [35] computing resources and the INFN Pisa HPC cluster Theocluster Zefiro.

# A  Kronecker tensor structures for complex fields

In this appendix we demonstrate how a product of two operators can be decomposed onto irreps of $U(m) \times U(n)$ in the picture where we work with complex fields. For simplicity we will start with just $U(n)$. The generalization to $U(m) \times U(n)$ is trivial. The main utility of working with complex fields is that the projectors take their simplest form possible, namely as combinations of Kronecker deltas with the least number of indices possible. The real field picture can also have its projectors expressed in terms of Kronecker deltas, albeit at the cost of more indices. We hope that presenting our projectors in three different pictures will make our work more intuitive. In order to capture all irreps that can appear in the real field notation, and hence not miss any information in the bootstrap algorithm, we need to consider two OPEs, namely $\Phi_i^\dagger \times \Phi_j$ and $\Phi_i \times \Phi_j$. Below we show how they may be decomposed onto irreps:

$$
\begin{aligned}
\Phi_i^\dagger \times \Phi_j &\sim \left( \Phi_i^\dagger \Phi_j - \frac{1}{n} \delta_{ij} \Phi_k^\dagger \Phi_k \right) + \frac{1}{n} \delta_{ij} \Phi_k^\dagger \Phi_k \,, \\
\Phi_i \times \Phi_j &\sim (\Phi_i \Phi_j + \Phi_j \Phi_i) + (\Phi_i \Phi_j - \Phi_j \Phi_i) \,.
\end{aligned}
\tag{A.1}
$$

The first line in (A.1) shows the decomposition into the adjoint ($R$) and singlet ($S$) representations, where as the second line shows the decomposition into the symmetric ($T$) and antisymmetric representations ($A$). To read off the projectors from (A.1) it is useful to remember

$$
O_{ij}^X \sim P_{ijkl}^X \Phi_k \Phi_l \,, \qquad O_{ij}^X \sim P_{ijkl}^X \Phi_k^\dagger \Phi_l \,,
\tag{A.2}
$$

where $O^X$ is the exchanged field in some irrep, e.g. $O_{12}^T \sim \Phi_1 \Phi_2 + \Phi_2 \Phi_1$. The first relation in (A.2) can be used to read off the $T$ and $A$ projectors, whereas the second can be used for $S$ and $R$. Notice that we have implicitly assumed that fields are inserted at different positions in order for antisymmetric irreps to not vanish identically. The projectors can be read off as[26]

$$
P_{ijkl}^S = \frac{1}{n} \delta_{ij} \delta_{kl} \,, \qquad P_{ijkl}^R = \delta_{ik} \delta_{jl} - \frac{1}{n} \delta_{ij} \delta_{kl} \,,
\tag{A.3}
$$

$$
P_{ijkl}^T = \frac{1}{2} (\delta_{ik} \delta_{jl} + \delta_{il} \delta_{jk}) \,, \qquad P_{ijkl}^A = \frac{1}{2} (\delta_{ik} \delta_{jl} - \delta_{il} \delta_{jk}) \,.
\tag{A.4}
$$

The dimensions of the corresponding irreps are $(1, (n-1)(n+1), \frac{1}{2}n(n+1), \frac{1}{2}n(n-1))$. As the reader may have observed from the main text, or the next appendix, when going from the complex field picture to the real field picture the above dimensions get multiplied by a factor of 2. For example $\frac{n(n+1)}{2}$ becomes $n(n+1)$. This is because each of the initial $\frac{n(n+1)}{2}$ complex elements contains two real elements. The last step now is to write down the projectors for $U(m) \times U(n)$. This is trivial, in the sense that they are just products of $U(m)$ with $U(n)$ projectors. We have

$$
\begin{aligned}
P_{ijklmnop}^S &= P_{ijkl}^S P_{mnop}^S \,, \quad P_{ijklmnop}^{RS} = P_{ijkl}^R P_{mnop}^S \,, \quad P_{ijklmnop}^{SR} = P_{ijkl}^S P_{mnop}^R \,, \\
P_{ijklmnop}^{RR} &= P_{ijkl}^R P_{mnop}^R \,, \quad P_{ijklmnop}^{TT} = P_{ijkl}^T P_{mnop}^T \,, \quad P_{ijklmnop}^{TA} = P_{ijkl}^T P_{mnop}^A \,, \\
P_{ijklmnop}^{AT} &= P_{ijkl}^A P_{mnop}^T \,, \quad P_{ijklmnop}^{AA} = P_{ijkl}^A P_{mnop}^A \,.
\end{aligned}
\tag{A.5}
$$

The sum rules that can be derived with the above projectors can be checked to be completely equivalent to those derived from the projectors of real fields outlined in the main text. Another observation is that, compared to real fields, we do not need separate projectors for even and odd spins.

---

[26]Note that we take the correlator to be $\langle \Phi_i^\dagger \Phi_j \Phi_k \Phi_l^\dagger \rangle$ which is why the projector of the adjoint representation is equal to $P_{ijkl}^R = \delta_{ik} \delta_{jl} - \frac{1}{n} \delta_{ij} \delta_{kl}$ instead of $P_{ijkl}^R = \delta_{il} \delta_{jk} - \frac{1}{n} \delta_{ij} \delta_{kl}$.

# B  Kronecker tensor structures for real fields

The projectors that correspond to a four-point function of real fields can be intuitively presented in terms of Kronecker deltas if we add an additional index. This form is useful since one may directly extract the form of exchanged operators, as we will show. The form of exchanged operators is useful to know since it can guide us with respect to assumptions we may impose. Also, we expect it to be easier to work with in a mixed correlator system. We start by labeling the real and complex parts of an operator $\Phi_i$ with an index (we start with $U(n)$ for simplicity)

$$\Phi_i = \phi_i^1 + i\,\phi_i^2 \,, \tag{B.1}$$

where the upper case $\Phi$ denotes the complex operator and the lower case $\phi$ denote real fields. We must now simply plug in (B.1) to the expressions for the representations of the previous appendix. For simplicity we will do this for the singlet representation, and then quote the results for rest of the representations. Note that implicitly we consider the two external fields of the OPE at different positions, for otherwise the antisymmetric combinations would vanish identically. We have

$$\Phi_i^\dagger \Phi_i = (\phi_i^1 \phi_i^1 + \phi_i^2 \phi_i^2) + i(\phi_i^1 \phi_i^2 - \phi_i^2 \phi_i^1) \,, \tag{B.2}$$

where the first parenthesis corresponds to what was called $S_{\text{even}}$ in the main text, and the second parenthesis corresponds to what was called $S_{\text{odd}}$. As expected $S_{\text{odd}}$ vanishes identically if we don't insert powers of derivatives between the operators. The projectors are now very straightforward to write down by recalling the relation

$$O_{ij;ab}^X = P_{ijkl;abcd}^X \phi_i^a \phi_j^b \,, \tag{B.3}$$

where $X$ stands for some specific irrep and indices from the beginning of the latin alphabet take the values $1,2$. Notice that (B.3) is simply the statement that projectors must project products of operators onto irreps. We have

$$
\begin{aligned}
P_{ijkl;abcd}^{S_{\text{even}}} &= \frac{1}{2n}\delta_{ab}\delta_{cd}\delta_{ij}\delta_{kl} \,, \\
P_{ijkl;abcd}^{S_{\text{odd}}} &= \frac{1}{2n}(\delta_{ac}\delta_{bd} - \delta_{ad}\delta_{bc})\delta_{ij}\delta_{kl} \,.
\end{aligned}
\tag{B.4}
$$

Indeed, one may confirm that, for example,

$$O_{11;11}^{S_{\text{even}}} \sim (\phi_i^1 \phi_i^1 + \phi_i^2 \phi_i^2) \sim P_{11kl;11cd}^{S_{\text{even}}} \phi_k^c \phi_l^d \,. \tag{B.5}$$

This procedure can be repeated for the rest of the irreps. The resulting projectors are

$$
\begin{aligned}
P_{ijkl;abcd}^{S_{\text{even}}} &= \frac{1}{2n}\delta_{ab}\delta_{cd}\delta_{ij}\delta_{kl} \,, \\
P_{ijkl;abcd}^{S_{\text{odd}}} &= \frac{1}{2n}(\delta_{ac}\delta_{bd} - \delta_{ad}\delta_{bc})\delta_{ij}\delta_{kl} \,, \\
P_{ijkl;abcd}^{R_{\text{even}}} &= \tfrac{1}{2}\delta_{ab}\delta_{cd}\Big(\delta_{ik}\delta_{jl} - \frac{1}{n}\delta_{ij}\delta_{kl}\Big) \,, \\
P_{ijkl;abcd}^{R_{\text{odd}}} &= \tfrac{1}{2}(\delta_{ac}\delta_{bd} - \delta_{ad}\delta_{bc})\Big(\delta_{ik}\delta_{jl} - \frac{1}{n}\delta_{ij}\delta_{kl}\Big) \,, \\
P_{ijkl;abcd}^{T_{\text{even}}} &= \tfrac{1}{4}(\delta_{ac}\delta_{bd} + \delta_{ad}\delta_{bc} - \delta_{ab}\delta_{cd})(\delta_{ik}\delta_{jl} + \delta_{il}\delta_{jl}) \,, \\
P_{ijkl;abcd}^{A_{\text{odd}}} &= \tfrac{1}{4}(\delta_{ac}\delta_{bd} + \delta_{ad}\delta_{bc} - \delta_{ab}\delta_{cd})(\delta_{ik}\delta_{jl} - \delta_{il}\delta_{jl}) \,.
\end{aligned}
\tag{B.6}
$$

The dimensions of the corresponding irreps are $(1, 1, (n-1)(n+1), (n-1)(n+1), n(n+1), n(n-1))$.

Using the above expressions, it is now trivial to write down the $U(m) \times U(n)$ projectors:

$$
\begin{aligned}
P^{S_{\text{even}}}_{ijklmnop;abcdefgh} &= P^{S_{\text{even}}}_{ijkl;abcd} P^{S_{\text{even}}}_{mnop;efgh} + P^{S_{\text{odd}}}_{ijkl;abcd} P^{S_{\text{odd}}}_{mnop;efgh}, \\
P^{S_{\text{odd}}}_{ijklmnop;abcdefgh} &= P^{S_{\text{even}}}_{ijkl;abcd} P^{S_{\text{odd}}}_{mnop;efgh} + P^{S_{\text{odd}}}_{ijkl;abcd} P^{S_{\text{even}}}_{mnop;efgh}, \\
P^{RS_{\text{even}}}_{ijklmnop;abcdefgh} &= P^{R_{\text{even}}}_{ijkl;abcd} P^{S_{\text{even}}}_{mnop;efgh} + P^{R_{\text{odd}}}_{ijkl;abcd} P^{S_{\text{odd}}}_{mnop;efgh}, \\
P^{RS_{\text{odd}}}_{ijklmnop;abcdefgh} &= P^{R_{\text{even}}}_{ijkl;abcd} P^{S_{\text{odd}}}_{mnop;efgh} + P^{R_{\text{odd}}}_{ijkl;abcd} P^{S_{\text{even}}}_{mnop;efgh}, \\
P^{SR_{\text{even}}}_{ijklmnop;abcdefgh} &= P^{S_{\text{even}}}_{ijkl;abcd} P^{R_{\text{even}}}_{mnop;efgh} + P^{S_{\text{odd}}}_{ijkl;abcd} P^{R_{\text{odd}}}_{mnop;efgh}, \\
P^{SR_{\text{odd}}}_{ijklmnop;abcdefgh} &= P^{S_{\text{even}}}_{ijkl;abcd} P^{R_{\text{odd}}}_{mnop;efgh} + P^{S_{\text{odd}}}_{ijkl;abcd} P^{R_{\text{even}}}_{mnop;efgh}, \\
P^{RR_{\text{even}}}_{ijklmnop;abcdefgh} &= P^{R_{\text{even}}}_{ijkl;abcd} P^{R_{\text{even}}}_{mnop;efgh} + P^{R_{\text{odd}}}_{ijkl;abcd} P^{R_{\text{odd}}}_{mnop;efgh}, \\
P^{RR_{\text{odd}}}_{ijklmnop;abcdefgh} &= P^{R_{\text{even}}}_{ijkl;abcd} P^{R_{\text{odd}}}_{mnop;efgh} + P^{R_{\text{odd}}}_{ijkl;abcd} P^{R_{\text{even}}}_{mnop;efgh}, \\
P^{TT_{\text{even}}}_{ijklmnop;abcdefgh} &= P^{T_{\text{even}}}_{ijkl;abcd} P^{T_{\text{even}}}_{mnop;efgh}, \\
P^{TA_{\text{odd}}}_{ijklmnop;abcdefgh} &= P^{T_{\text{even}}}_{ijkl;abcd} P^{A_{\text{odd}}}_{mnop;efgh}, \\
P^{AT_{\text{odd}}}_{ijklmnop;abcdefgh} &= P^{A_{\text{odd}}}_{ijkl;abcd} P^{T_{\text{even}}}_{mnop;efgh}, \\
P^{AA_{\text{even}}}_{ijklmnop;abcdefgh} &= P^{A_{\text{odd}}}_{ijkl;abcd} P^{A_{\text{odd}}}_{mnop;efgh}.
\end{aligned}
\tag{B.7}
$$

From these expressions we can also see explicitly that when $m = n$, if we choose to consider the two $U(n)$ symmetries as indistinguishable (which we remind the reader is not strictly necessary), the *RS* irreps become the same as the *SR* irreps. The same also happens for *TA* and *AT*.

## C  Numerical parameters

For most of our plots, the bounds are obtained with the use of PyCFTBoot [30] and SDPB [38]. We use the numerical parameters m_max = 6, n_max = 9, k_max = 36 in PyCFTBoot, and we include spins up to l_max = 26. The binary precision for the produced xml files is 896 digits. SDPB is run with the options --precision=896, --detectPrimalFeasibleJump, --detectDualFeasibleJump and default values for other parameters. We refer to this set of parameters as "$A$". Unless otherwise stated, our plots are run with parameters "$A$".

For some of the plots we used m_max = 5, n_max = 7, l_max = 36, k_max = 42 and m_max = 6, n_max = 9, l_max = 36, k_max = 42, referred to as "$B$" and "$C$" respectively. Lastly, we also used qboot [39], with $\Lambda = 15$, n_max = 500, $\nu$_max = 25 and $l = \{0\text{–}49, 55, 56, 59, 60, 64, 65, 69, 70, 74, 75,$
$79, 80, 84, 85, 89, 90\}$ referred to as "$D$".

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
