# Peer review of "CFTs with $U(m)\times U(n)$ Global Symmetry in 3D and the Chiral Phase Transition of QCD"

_SciPost Physics, doi:SciPost Phys. 15, 075 (2023)_

## Round 1 · Referee Report · Anonymous (Referee 1) · 2022-12-19

Report

The paper under review studies $SU(N) \times SU(M) \times U(1)$ symmetric CFTs from the conformal bootstrap approach. While the methods used in the paper are not brand new and the direct physical implication in nature is not immediately obvious (except for $N=M = 2$ case where there already have been other studies), the paper gives some addition to our knowledge of three-dimensional CFTs.

Before publication, I think several clarifications, as well as new additional information, are needed.

(1) On page 25, it was mentioned "QCD with 1000 massless flavors", but we usually call $SU(3)$ gauge theory QCD, and with 1000 flavors, it would not be confining. I can easily guess that what is meant here was larger color Yang-Mills theory with 1000 massless flavors, but more careful rewriting seems welcomed.

(2) The intrinsic difference between $U_+$ and $U_-$ is the stability (as reviewed in section 2 of the paper). It seems extremely important to study the operator contents at various "kinks" to see whether we have one or two relevant singlet scalar operator(s) in the spectrum. Since the authors seem to have an available method (i.e. extremal functional technique) from the previous studies, I strongly recommend mentioning the stability of these potential fixed points (kinks). In particular, any potential fixed points corresponding to QCD phase transition should have only one singlet relevant operator.

(3) As we know, some of the kinks can be interpreted as more symmetric CFTs (e.g. O(2NM) fixed point). I wonder if the kink found in $SU(3) \times SU(3) \times U(1)$ in this paper can be interpreted as $SU(L)$ or $U(L)$ fixed point. It seems that plot 12 of this paper looks very similar to plot 1 of
https://arxiv.org/abs/1705.02744 (where $SU(6)$ adjoint plot was shown).
If this is the case, the nature of the fixed point is not of $SU(3) \times SU(3) \times U(1)$ but it has the enhanced symmetry: then it has little to do with the QCD phase transition.

---

## Round 1 · Referee Report · Anonymous (Referee 2) · 2022-12-25

Report

This work revisits the prospect of bootstrapping chiral symmetry breaking for 3d nuclear matter at finite temperature. To a good approximation, this leads one to consider a scalar order parameter (which can be thought of as a quark bilinear) in the bifundamental of $U(m) \times U(n)$. The most physical cases are $m = n = 2, 3$ but the authors take a more general point of view, also studying regimes where one or both parameters are large. The perturbative sections of this paper, which assume the transition is reachable from a standard Landau-Ginzburg action, estimate the critical values of $m$ and $n$ at which it becomes first order. This is followed by a numerical bootstrap section which supports this basic picture and reveals a surprising tendency for certain fixed points to saturate the bounds even when there is no kink. For both types of calculations, the authors show how the necessary invariant theory is made more straightforward by embedding $U(m) \times U(n)$ into $O(2mn)$.

This is a well written paper which finds new results about an important phenomenological problem. I have only minor changes to recommend.

Requested changes

  1. On page 10, it might be good to quickly say that the notation $RSSR$ for the sum will be used later.
  2. The caption of Figure 11 should only mention circles and squares for $n = 10, 20$.
  3. I think removing "number of" would make the first sentence of the last paragraph of section 7 sound better.
  4. Appendix C's "plots are ran" should say "plots are run".
  5. On page 6-7, I think it is confusing to say there are 9 rank four invariant tensors for $U(m) \times U(n)$ and 7 for indistinguishable $U(n)^2$. The counting here should match the number of projectors: 12 and 9. The reduction in number appears to come from treating index permutations of $\omega_{u, ijkl}$ as equivalent. But then it seems inconsistent to not also do this for $\delta_{ij} \delta_{kl}$.

---

## Round 2 · Referee Report · Anonymous (Referee 2) · 2023-2-3

Report

I thank the authors for their time updating the paper. It should now proceed to publication.

---

## Round 2 · Referee Report · Anonymous (Referee 3) · 2023-2-6

Strengths

This is an interesting paper exploring bootstrap and epsilon expansion bounds
which are claimed to have some possible applications to QCD at finite
temperature.

Apart from some important but fixable (at a cost of changes to e.g. the title
abstract, and introduction) issues to do with the desired application to QCD, the body of
the paper paper seems well-written and technically strong.

Weaknesses

I am confused by some statements in the introduction concerning U(1)A symmetry. As the authors note, due to the ABJ anomaly there is no such global symmetry in QCD. As their references 1-4 show, in the limit T/\Lambda \to \infty, the instanton density approaches to zero, the ABJ anomaly becomes negligible, and U(1)A becomes a good approximate symmetry of the resulting high-temperature EFT. But I'm not aware of any plausible argument that the instanton density drops to exactly zero at T_c, given that T_c/\Lambda_QCD \sim O(1). Without such an argument, U(1)_A is not restored for any finite value of T/\Lambda, and in particular it is not restored at T_c. So I don't understand why G_{LRA} would be relevant for the thermal chiral phase transition in three-color QCD. Can the authors supply an argument on why they think U(1)_A should be restored at T_c?

As noted at the top of page 3 restoration of U(1)_A at T_c is crucial for there to be an application of the results of this paper to QCD. Given that I think there are persuasive reasons to think U(1)_A is NOT restored at T_c, is there some reason the authors did not explore bootstrap founds on SU(N) \times SU(N) 3d CFTs? If there's some technical reason for this it should be discussed in the paper.

Report

As explained above I don't think this paper can say something for three-color
QCD with any number of massless quarks. However, the way to rescue the
phenomenogical viability of the choice to consider U_1(A) is via the 1/N_c
expansion. That is, instanton effects are also suppressed by 1/N_c for any
temperature, and U(1)_A is expected to be restored in large-N_c QCD. Therefore
I think that the results of this paper help deduce constraints on N_f = 2 and
N_f = 3 QCD *in the large N_c limit*. If the authors agree, they should make this clear in all the places they refer to QCD, such as such as even the title. (I am very fond of large N
QCD, but large N QCD \neq QCD as such.).
  • validity: high
  • significance: good
  • originality: good
  • clarity: good
  • formatting: excellent
  • grammar: perfect

Author:  Stefanos Robert Kousvos  on 2023-03-06  [id 3440]

(in reply to Report 2 on 2023-02-06)

We thank the referee for a careful reading of our manuscript, as well as pointing out possible issues with our discussion in the introduction.

Let us note that our numerical bounds, due to the conformal bootstrap, do indeed also apply to $SU(n) \times SU(n)$ 3d CFTs. This is because the sum rules for $U(n) \times U(n)$ and $SU(n) \times SU(n)$ resulting from a four point function of operators in the defining representation are identical (with one particular exception, n=4, now mentioned in the introduction). Thus, our numerical results do indeed apply even if the axial symmetry is not restored exactly.

On the other hand, our perturbative results only apply to the case where the axial symmetry is restored. This is because of the term in the Lagrangian corresponding to the breaking of $U(1)_A$, namely $\delta \mathcal{L}\sim g(det(\Phi)+ det(\Phi^\dagger))$ ($g$ being a coupling). The perturbative fixed points in our work are reached under the necessary condition $g=0$.

To clarify the possible applicability of our results to QCD, we have added a new paragraph close to the top of page 3. Additionally, we made a number of adjustments to phrases in the introduction as well as the numerical bootstrap results section to reflect this. We would be happy to clarify further if the referee believes there are additional issues left un-adressed.

Attachment:

CFTs_with_U_m_xU_n__Global_Symmetry_in_3D_and_the_Chiral_Pha_SVoBklu.pdf

---

## Round 2 · Referee Report · Anonymous (Referee 1) · 2023-2-16

Report

In my opinion, conformal bootstrap is closer to experimental physics, and it does not seem scientifically sound that the authors report only results that are favorable to their interpretation while hiding the results that are not (i.e. the mismatch of the spectrum or stability condition in this case). I will eventually leave the editor on the decision, but if I were the authors, I would comment on the discrepancies in the paper.
  • validity: -
  • significance: -
  • originality: -
  • clarity: -
  • formatting: -
  • grammar: -

Author:  Stefanos Robert Kousvos  on 2023-03-06  [id 3439]

(in reply to Report 3 on 2023-02-16)

We have added footnote 19 which comments on the absence of the "$\phi^4$" singlet from the spectrum. The footnote also gives examples of other work where operators have been found to be missing from the spectrum ($O(m)\times O(n)$ symmetric theories and the Ising model).

Attachment:

CFTs_with_U_m_xU_n__Global_Symmetry_in_3D_and_the_Chiral_Pha_VdwCqTT.pdf

---

## Round 2 · Author Response

Below we address the issues raised by the referees. We hope our modifications sufficiently address their concerns.

---

## Round 2 · List of Changes

Report 2:

We thank the referee for carefully reading through our manuscript and for pointing out a number of misprints. We have corrected them accordingly. Additionally, we corrected a misidentification of $U_+$ and $U_-$ in parts of the text.

Report 1:

We thank the referee for their careful reading of our manuscript, especially for pointing out possible bound coincidences with earlier work.

Regarding said bound coincidences: for two bootstrap systems to have bound coincidences, the group theoretic dimensions of the externals must agree. For example, the bootstrap of a four point function of traceless symmetric two-index tensors of $O(n)$, $t_{ij}$, has bounds that coincide with the bootstrap of a four point function consisting of four $O(N)$ vectors $\phi_i$ with $N=(n-1)(n+2)/2$, i.e. the group theoretic dimension of $t_{ij}$. Similarly, a bootstrap system of $U(m)\times U(n)$ operators in the defining representation can enhance to $O(N)$ with $N=2mn$, i.e. the dimension of the defining representation in $U(m)\times U(n)$.

With these observations, we see that the $SU(6)$ adjoint bootstrap in the reference by Nakayama cannot directly coincide with our $U(3)\times U(3)$ bootstrap. This is because the group-theoretic dimension of the SU(6) adjoint is $6^2-1=35$, whereas the group-theoretic dimension of the external operator in our case is $3\cdot 3=9$.

Regarding the extraction of the extremal spectra, we did extract some sample spectra, however these do not seem to give conclusive results. For example, extracting the spectrum corresponding to $U_-$ for $U(2)\times U(20)$ (which should be unstable as a fixed point), we found one operator with dimension close to $\Delta_S=1$ and one operator with dimension close to $\Delta_S =3$, corresponding in the field theory language to "$\phi^2$" and "$\phi^6$". In other words the extremal spectrum misses the "$\phi^4$" singlet operator. We observed a similar situation when studying $O(m)\times O(n)$ symmetric theories in previous work. Hence we opted to not argue about stability based on extremal spectra.

Let us note that we also fixed a mislabeling of $U_+$ and $U_-$ in the text. $U_+$ (the stable fixed point) should be the fixed point with two Hubbard-Stratonovic fields at large-n, and $U_-$ the fixed point with one Hubbard-Stratonovich field at large-n.

Lastly, we added a footnote on page 25 pointing out that one would typically need a sufficiently large number of colors for the theory to be confining.

---

## Round 3 · Referee Report · Anonymous (Referee 2) · 2023-4-5

Report

This paper looks like it's ready for publication.

I am not bothered by the singlets which could not be found using the extremal functional. It has been known for awhile that this method, with achievable computing power, generically misses several operators even at low scaling dimension.

---

## Round 3 · Referee Report · Anonymous (Referee 1) · 2023-4-6

Report

As for my request, the revision seems satisfactory and I suggest the manuscript be accepted.

As for the comment by the other referee, I think that the large N QCD may not be an ideal place to realize the fixed points studied in this manuscript. I believe that the large N QCD shows the first-order phase transition rather than the second-order phase transition due to the jump of free energy (order N^2 versus order N^0), which is also suggested by holographic constructions.

---

## Round 3 · List of Changes

Below we list the changes made to address the issues raised by each report.

Report 3:

We have added footnote 19 which comments on the absense of the "$\phi^4$" singlet from the spectrum. The footnote also gives examples of other work where operators have been found to be missing from the spectrum ($O(m)\times O(n)$ symmetric theories and the Ising model).

Report 2:

We thank the referee for a careful reading of our manuscript, as well as pointing out possible issues with our discussion in the introduction.

Let us note that our numerical bounds, due to the conformal bootstrap, do indeed also apply to $SU(n) \times SU(n)$ 3d CFTs. This is because the sum rules for $U(n) \times U(n)$ and $SU(n) \times SU(n)$ resulting from a four point function of operators in the defining representation are identical (with one particular exception, n=4, now mentioned in the introduction). Thus, our numerical results do indeed apply even if the axial symmetry is not restored exactly.

On the other hand, our perturbative results only apply to the case where the axial symmetry is restored. This is because the term in the Lagrangian corresponding to the breaking of $U(1)_A$, namely $\delta \mathcal{L}\sim g(det(\Phi)+ det(\Phi^\dagger))$ ($g$ being a coupling). The perturbative fixed points in our work are reached under the necessary condition $g=0$.

To clarify the possible applicability of our results to QCD, we have added a new paragraph on the top of page 3. Additionally, we made a number of adjustments to phrases in the introduction as well as the numerical bootstrap results section to reflect this. We would be happy to clarify further if the referee believes there are additional issues left un-adressed.

---

## Editorial Decision

published